

# Changes in the impacts of ship emissions on PM$_{2.5}$ and its components in China under the staged fuel oil policies

Guangyuan Yu[1,2], Yan Zhang[1,3,4*], Qian Wang[2], Zimin Han[1], Shenglan Jiang[1], Fan Yang[5], Xin Yang[6], and Cheng Huang[2*]

[1]Shanghai Key Laboratory of Atmospheric Particle Pollution and Prevention (LAP[3]), National Observations and Research Station for Wetland Ecosystems of the Yangtze Estuary, Department of Environmental Science and Engineering, Fudan University, Shanghai 200438, China

[2]Shanghai Environmental Monitoring Center (SEMC), Shanghai 200235, China

[3]Shanghai Institute of Eco Chongming (SIEC), Shanghai 200062, China

[4]MOE laboratory for National Development and Intelligent Governance, Shanghai institute for energy and carbon neutrality strategy, IRDR ICoE on Risk Interconnectivity and Governance on Weather/Climate Extremes Impact and Public Health, Fudan University, Shanghai 200433, China

[5]Pudong New Area Environmental Monitoring Station, Shanghai 200135, China

[6]Shenzhen Key Laboratory of Precision Measurement and Early Warning Technology for Urban Environmental Health Risks, School of Environmental Science and Engineering, Southern University of Science and Technology, Shenzhen 518055, China

*Correspondence to*: Yan Zhang (yan_zhang@fudan.edu.cn) and Cheng Huang (huangc@saes.sh.cn)

**Abstract.** The issue of air pollution caused by ship emissions is becoming prominent with the increasing global shipping activities. China has carried out staged fuel oil policies in the past few years to meet the requirements of the global low sulfur regulation by the International Marine Organization (IMO), called the IMO Regulation. However, the impacts of ship emissions on air quality in China after 2020 are not sufficiently understood. This study firstly updated the ship emission inventory including PM$_{2.5}$ components based on field and on-board measurements under the staged fuel oil policies. Then, the impacts of ship emissions on PM$_{2.5}$ as well as its gas precursors and primary and secondary components in China from 2017 to 2021 have been revealed by using the Weather Research and Forecasting (WRF) model and the Community Multi-scale Air Quality (CMAQ) model. We found that ship emissions increased the PM$_{2.5}$ concentrations up to 3.8 μg m$^{-3}$ in 2017 and 2.6 μg m$^{-3}$ in 2021 along China's coastal area. The areas with high concentration levels widely distributed over the offshore waters in 2017, and shrunk to some parts of China's coast in 2021. The contributions of ship emissions to the PM$_{2.5}$ concentrations over China's main port cities ranged from 3.0% to 17.4% in 2017 and 2.5% to 10.3% in 2021. Our findings suggest that it is important to consider both transport pathways and formation mechanisms of secondary aerosols to combat the PM$_{2.5}$ pollution caused by shipping in different regions.



## 1 Introduction

Shipping is the backbone of global trade and transports more than 80% of global goods. The global

shipping activities increased by ~20% in the past decade, and continues to grow at a rate of ~2% per year

in the coming years (UNCTAD, 2023). Meanwhile, heavy fuel oil (HFO) is the most widely used type

of fuels for marine vessels. The combustion of HFO can release remarkably higher amounts of sulfur

oxides ($SO_x$), particulate matter (PM), and trace elements compared to the combustion of lighter oils

(Agrawal et al., 2008a; Moldanová et al., 2013; Zhang et al., 2019b). Due to the increasing shipping

activities and the low quality of fuel oils, shipping is becoming an important source of air pollution,

especially in coastal areas and ports with dense traffic (Dalsøren et al., 2009; Eyring et al., 2010). It is

reported that international vessels emitted 9600 kt (kilotons) $SO_x$, 17100 kt $NO_x$ (nitrogen oxides), and

1351 kt $PM_{2.5}$ (PM with a diameter of less than 2.5 μm) in 2018 (IMO, 2021). In Europe, the maritime

transport sector produced 24% of all $NO_x$ emissions, 24% of all $SO_x$ emissions, and 9% of all $PM_{2.5}$

emissions in 2018, affecting ~40% of Europeans living within 50 km of the sea (EMSA and EEA, 2021).

In 2022, China held a national port cargo throughput of 15.68 billion tons, and was home to eight of the

top ten ports for cargo throughput and seven of the top ten ports for container throughput worldwide

(Ministry of Transport of the People's Republic of China, 2023). Tracking ship emissions and their

environmental impacts in China is of great significance.

Exposure to high levels of $PM_{2.5}$ can increase health problems like respiratory and cardiovascular

diseases. The World Health Organization (WHO) Global Air Quality Guidelines 2021 recommend that

annual mean concentrations of $PM_{2.5}$ should not exceed 5 μg/m³ (WHO, 2021). Studies using chemical

transport models (CTMs) have been conducted to simulate the impact of ship emissions on $PM_{2.5}$ in the

regions with heavy ship traffic. In China, ship emissions increased the annual averaged $PM_{2.5}$

concentrations up to 5.2 μg m⁻³ in 2015, surpassing the limit value supposed by the WHO. In Europe and

North America, the increase in $PM_{2.5}$ concentrations due to shipping is generally less than 2 μg m⁻³;

however, its relative contribution is significant, reaching 25%−50% along main shipping routes and

12%−15% in coastal areas (Aksoyoglu et al., 2016; Tang et al., 2020; Fink et al., 2023a; Golbazi and

Archer, 2023). Based on observation, the impact of ship emissions on PM can also be calculated by using

source apportionment methods like receptor models. In China, our previous studies show that ship

emissions contribute 1.96 μg m⁻³ (4.23%) to the ambient $PM_{2.5}$ concentration at port, while 0.4−3.1 μg



m$^{-3}$ (1.3%−8.8%) for downtown Shanghai (Zhao et al., 2013; Yu et al., 2021); the fraction of shipping-related particles is 1%−10% in port cites (Liu et al., 2017b; Wang et al., 2019; Zhang et al., 2019a; Zhai et al., 2023). Ship emissions contribute to annual mean concentrations of PM$_{2.5}$ with 1%−14% in

European coastal areas and 3%−9% in American coastal areas (Agrawal et al., 2009; Viana et al., 2014; Kotchenruther, 2015; Anastasopolos et al., 2021).

Shipping-related PM is comprised of primary particles and secondary products. Ships primarily emit organic carbon (OC), elemental carbon (EC), sulfate, metallic elements, etc., among which OC and sulfate are the main components of primary PM from ships burning HFO (Agrawal et al., 2008b; Lack

et al., 2009; Agrawal et al., 2010; Huang et al., 2018a; Yang et al., 2022; Karjalainen et al., 2022). Vanadium (V), nickel (Ni), and the V/Ni ratio are the mostly used tracers of ship emissions (Agrawal et al., 2009; Moldanová et al., 2009; Celo et al., 2015; Corbin et al., 2018; Yu et al., 2021). Calculating the emissions, the concentrations, and the deposition fluxes of V and Ni from shipping can help better understanding their geochemical cycles. In comparison, sulfate, nitrate, and ammonium (SNA) dominate

the shipping-related PM with its proportion even exceeding 90% based on modeling research (Lv et al., 2018; Jonson et al., 2020; Fink et al., 2023a; Jang et al., 2023).

Studies have demonstrated that using desulfurized fuel oils can significantly reduce the emissions of various air pollutants such as SO$_x$, PM, OC, heavy metals, and polycyclic aromatic hydrocarbons (PAHs) (Tao et al., 2013; Zetterdahl et al., 2016; Kotchenruther, 2017; Spada et al., 2018; Huang et al., 2018a).

To combat the air pollution caused by ship emissions, four Emission Control Areas (ECAs) have been established in Europe and North America since 2011. The sulfur limits for fuel in the ECAs was restricted to 0.10% m/m after 1 January 2015. Besides, the Tier III entered into force on 1 January 2016 in the ECAs of North America and on 1 January 2021 in the ECA areas of Europe. However, the regulations in China are significantly lagging behind those in North America and Europe. Ships berthing at the ports in

the China's Domestic Control Areas (DECAs) were required to use fuel with a sulfur content no more than 0.50% (low sulfur fuel oil, LSFO hereafter) after 1 January 2017, which is called the DECA 1.0 period. All ships within 12 nm (nautical miles) from the coastline must use LSFO after 1 January 2019, which is referred to as the DECA 2.0 period (Liu et al., 2018a; Wang et al., 2021). The inland emission control areas covering the Yangtze River, the Xijiang River, and the Pearl River went into effect after 1

January 2019 where coastal vessels were required to combust LSFO; both coastal and international vessels must use fuel with a sulfur limit of 0.10% called ultra-low sulfur fuel oil (ULSFO) after 1 January



2020. The rapid transition of China's control measures is to meet the global fuel sulfur limit of 0.50% (reduced from 3.50%) from 1 January 2020 mandated by the IMO (the IMO Regulation). For the $NO_x$ emission control, newly built ships in China follows the IMO Tier II standard from 2011. The staged

sulfur regulations are expected to effectively alleviate the air pollution from shipping in China. However, there are very few studies on the impacts of ship emissions on air quality after the implementation of the IMO Regulation based on actual shipping activity data and CTMs (Zhai et al., 2023; Feng et al., 2023). For instance, the simulation years of the recently published studies from China, Europe, and North America are generally before the IMO Regulation (Fink et al., 2023b; Fu et al., 2023; Golbazi and Archer,

2023). There is relatively little research on simulating the specific composition of shipping-related PM. The impacts of meteorology and chemical mechanisms on the $PM_{2.5}$ pollution caused by shipping in China are not fully understood. In addition, it has been observed that the concentrations of V and Ni from shipping experienced a markedly and stepwise decrease in China's largest port city from 2017 to 2020 in our previous study. The latest emission inventories of V and Ni from shipping are still not earlier than

2017, and need updating until after 2020 (Zhao et al., 2021; Jiang et al., 2024).

     In this study, we updated the ship emission inventory based on the data from the Automatic Identification System (AIS), and simulated the impacts on $PM_{2.5}$ in China as well as its gas precursors ($SO_2$ and $NO_2$) and components from 2017 to 2021 by using the Weather Research and Forecasting (WRF) model and the Community Multi-scale Air Quality (CMAQ) model. The emissions of V and Ni from

shipping were constrained by the field observational data from our previous study and the results of on-board emission measurements. Based on the simulation results, the spatiotemporal patterns of shipping-related $PM_{2.5}$ as well as trace elements (V and Ni), secondary inorganic aerosols, and organic aerosols were obtained. Meanwhile, the interannual and seasonal variations of the impacts were investigated. Then, we focused on the changes in the impacts due to the implementation of the IMO Regulation at the port

city level. Besides, the roles of the meteorological factors in affecting the seasonal and diurnal patterns of primary PM from shipping over the port cities were discussed.



## 2 Methods

### 2.1 Setup of the WRF/CMAQ

We utilized the CMAQ version 5.4 to simulate the pollutant concentrations, and the WRF version 4.1.1 to provide the meteorological input fields for the CMAQ. The WRF physics scheme configuration included the Yonsei University (YSU) for the planetary boundary layer (PBL) scheme, the Noah land surface scheme, the Thompson microphysics scheme, the rapid radiative transfer model for general circulation models (RRTMG) for short and long wave schemes, and the Kain-Fritsch cumulus scheme

for cumulus parameterization. 40 vertical layers were setup with the model top pressure at 50 hPa, among which 12 layers were distributed within 1.6 km above the surface. The surface layer thickness was ~50 m. The WRF model was driven by the European Centre for Medium-Range Weather Forecasts (ECMWF) Reanalysis v5 (ERA5) at hourly temporal and $0.25° \times 0.25°$ spatial resolution (Hersbach et al., 2023). To reduce the errors, monthly WRF simulations were divided into six runs, each of which included 12-h

spin-up time. Two nested domain simulations were operated with horizontal resolutions of 27 km × 27 km (d01) and 9 km × 9 km (d02) encompassing East Asia and eastern China, respectively. In the CMAQ model, two grids on each WRF lateral boundary were removed, and thus there were 161 × 174 and 233 × 215 grids in d01 and d02, respectively. Figure 1 shows the nested domains configured in the CMAQ as well as the coastal emission control area (CECA) of China covering all the marine waters within 12

nm beyond the territorial baselines.

    The CMAQ model was configured to the gas-phase mechanism of Carbon Bond 6 revision 5 (CB6r5) and the aerosol module of AERO7. By modifying the aerosol module and the in-line dust module, two trace elements, V and Ni, were added into the CMAQ as inert aerosol components which only participate in atmospheric physical processes such as diffusion, advection, and deposition. Detailed information on

the code modification can be found in our previous study (Jiang et al., 2024). The initial and boundary conditions for d01 originated from the seasonal average hemispheric CMAQ output from the Community Modeling and Analysis System (CMAS) data repository, while those for d02 were derived from the output data of d01. For the analysis of seasonal variations, the simulations were conducted for January, April, July, and October of 2017 and 2021, representing winter, spring, summer, and autumn, respectively.

The annual average was equal to the average of four representative months. To study the impacts under staged fuel oil policies and save computing resources meanwhile, the simulations were operated for each



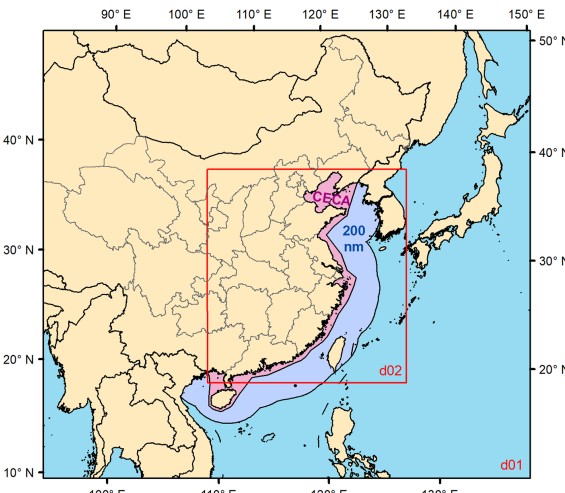

**Figure 1.** Map of the nested domains configured in the CMAQ. The coastal emission control area (CECA) of China is colored in pink. The boarder of the area within 200 nautical miles (nm) from the coastline of Chinese Mainland is outlined.

**Table 1.** Details of the WRF/CMAQ configuration.

| | |
|---|---|
| Simulation period | January, April, July, and October of 2017 and 2021; April of 2018–2020 |
| Grid resolution | 27 km × 27 km (d01), 9 km × 9 km (d02) |
| Vertical layers | 40 |
| Surface layer thickness | 50 m |
| Top of model | 50 hPa |
| WRF version 4.1.1 | |
| Grid size | 165 (south to north) × 178 (west to east) (d01); 237 × 219 (d02) |
| Initial/boundary conditions | ERA5 (ECMWF Reanalysis v5), hourly, 0.25° × 0.25° |
| Microphysics scheme | Thompson |
| Land surface model | Noah |
| Planetary boundary layer scheme | YSU (Yonsei University), topo_wind = 2 |
| Cumulus scheme | Kain-Fritsch |
| Shortwave radiation | RRTMG (Rapid Radiative Transfer Model for General circulation models) |
| Longwave radiation | RRTMG |
| Spin-up time | 12 h |
| Number of days per run | 6.5* |
| CMAQ version 5.4 | |
| Grid size | 161 × 174 (d01); 233 × 215 (d02) |
| Initial/boundary conditions | EPA 2017–2018 (d01); output of d01 run (d02) |
| Gas-phase mechanism | Carbon Bond 6, revision 5 (CB6r5) |
| Aerosol module | AERO7 |
| Spin-up time | 5 days |
| Number of days per run | 36 for January, July, and October; 35 for April |

*For WRF, the last run of April is an exception with the time length of 5.5 days.

April from 2017 to 2021 based on our previous finding that the impacts of ship emissions in China's coastal areas usually peak in spring (Yu et al., 2021). The spin-up time of each simulation was 5 days.

Detailed information on the WRF/CMAQ configuration can be seen in Table 1. The impacts of ship



emissions were extracted based on the zero-out method, i.e., two runs with and without ship emissions, named the base run and the exship run respectively in this study.

## 2.2 Emission data

### 2.2.1 Ship emissions

A bottom-up ship emission model based on the AIS data was used to calculate the emission inventories of $SO_2$ (sulfur dioxide), $NO_x$, CO (carbon monoxide), NMVOCs (nonmethane volatile organic compounds, $PM_{10}$ (PM with a diameter less than 10 μm), $PM_{2.5}$, $NH_3$ (ammonia), V, and Ni. Detailed information on the setup of the ship emission model and the low load adjustment multipliers for main engine emission factors (EFs) can be found in the previous studies (Fan et al., 2016; Feng et al., 2019). In the ship emission model, the power-based EFs under the staged fuel oil policies are categorized by engine type (Table S1).

During the DECA 1.0 period (2017−2018), we adopted the EFs of various species from Fan et al. (2016) and the fourth IMO Greenhouse Gas (GHG) study. The default setting for main engines (MEs) including slow-speed diesel (SSD) and medium-speed diesel (MSD) engines usually installed for large vessels was using high sulfur fuel oil (HSFO) with a sulfur content of ~2.7%. A high-speed diesel engine as a main engine (ME_HSD) generally used marine diesel oil (MDO) or marine gas oil (MGO) with a sulfur content of ~0.1%. An auxiliary engine (AE) was assumed to use LSFO with a sulfur content of ~0.5%.

During the DECA 2.0 period (2019), the settings remained the same as those during the DECA 1.0 period in the marine areas outside the CECA. In the CECA, the scenario of EF setting for MEs of large vessels was using LSFO. For the AE, considering that LSFO is often used by sea-going vessels in addition to ULSFO, we took the mean values of the EFs for LSFO and ULSFO.

After the implementation of the IMO Regulation, the settings in all marine areas followed those in the CECA during the DECA 2.0 period. Over land areas, for the sea-going vessels, the EFs of $SO_2$ and PM for ME were scaled by a factor of 0.2 and 0.26, and 0.315 and 0.391 for AE, respectively, which is due to the implementation of the inland river emission control areas. The EFs of V and Ni for ships navigating in inland waters were lowered by a factor of 10 for the ME due to the nonlinearity between the contents of sulfur and trace elements.



In terms of the EFs of V and Ni for marine vessels during the DECA 1.0 period, we used the values reported in the literature listed in Table S2. These values during the DECA 2.0 and after 2020 were reduced corresponding to the change ratios reported in our previous study (Yu et al., 2021). The emission inventories of V and Ni were validated through comparison between simulation results and observational data in Shanghai.

The mapping of $PM_{2.5}$ components from shipping to the AERO7 species is shown in Table S3. The mass fractions before the IMO Regulation (2017−2019) and after 2020 were referenced from Huang et al. (2018a) and Yang et al. (2022), respectively. The convert factors of NMVOCs emissions from shipping to lumped species in the CB6 mechanism were based on the median VOC profiles from the literature (Table S4) (Agrawal et al., 2008a; Huang et al., 2018b; Zhang et al., 2024). The initial height of ship

emissions considering plume rise and diffusion is basically with the range of 20−100 m (Chosson et al., 2008; He et al., 2021; Badeke et al., 2022; Lansø et al., 2023). We allocated 20% of the sea-going vessel emissions to the surface layer (0−50 m) and 80% to the second layer (50−109 m), while all the inland ship emissions were assigned in the surface layer (Table S5).

### 2.2.2 Land-based emissions

For land-based anthropogenic emissions, we used the Multiresolution Emission Inventory for China (MEIC) in 2017−2020 for mainland China and the MIX emission data in 2010 for East Asia excluding mainland China (Li et al., 2017; Zheng et al., 2021). The $NH_3$ emissions in the MEIC were replaced by the PKU-$NH_3$ inventory in 2017 (Kang et al., 2016). For natural sources, we used the CAMS-GLOB-BIO v3.1 for monthly global biogenic VOC (BVOC) emissions in 2017−2021 and the biogenic emissions

inventory from urban green spaces in China (OUC-BUGS) in 2017−2019 (Sindelarova et al., 2021; Ma et al., 2022). The MEIC, MIX, PKU-$NH_3$, and OUC-BUGS inventories were downloaded from the website (http://meicmodel.org.cn/). The grid resolutions of MEIC/MIX, BVOC, UBVOC, and PKU-$NH_3$ are 0.25°, 0.25°, 27 km, and 0.1°, respectively. The $PM_{2.5}$ profiles in Liu et al. (2017a) were used to convert $PM_{2.5}$ from non-shipping emissions to the AERO7 species. We multiplied the $PM_{2.5}$ emissions

from the MEIC/MIX by the V and Ni fractions in $PM_{2.5}$ in China by source to obtain the V and Ni emissions from anthropogenic sources excluding shipping (Liu et al., 2018b). The source-specific vertical profiles for industrial, power, residential, and land-based transportation emissions in Table S5 were referenced from Zheng et al. (2019).



### 2.3 Observational data and evaluation of simulation results

The observational data from the national meteorological stations and the national air quality monitoring stations listed in Table S6 were used to evaluate the simulation results. The hourly meteorological data were downloaded from the website (http://data.cma.cn), and the hourly air quality data were obtained from China National Environmental Monitoring Center (https://air.cnemc.cn:18014). To evaluate the model performance for $PM_{2.5}$ in coastal areas, 21 port cities along the coast of China were

selected as representatives. They rank among the top 20 in terms of cargo or container throughput nationwide and distribute on the coasts of the Bohai Rim (Dalian, Yingkou, Caofeidian, Binhai of Tianjin, and Yantai), the Yellow Sea (Qingdao, Rizhao, and Lianyungang), the Yangtze River Delta (Shanghai, Ningbo, Zhoushan, Hangzhou, Nantong, Zhangjiagang, and Nanjing), the Pearl River Delta (Shenzhen, Guangzhou, and Zhuhai), and the Beibu Gulf (Qinzhou) as well as on the west coast of the Taiwan Strait

(Fuzhou and Xiamen). The simulated concentrations in the surface layer were used to compare with the observational data.

       We used the hourly observational data from the Pudong site of Shanghai to evaluate the model performance of the tracers of ship emissions (V and Ni) and the secondary inorganic aerosols ($SO_4^{2-}$, $NO_3^-$, and $NH_4^+$). The metallic elements and the ions were measured by the Model Xact 625 (Cooper

Environmental Services, LLT, OR, USA) and the MARGA (Model ADI 2080, Applikon Analytical B. V. Corp., The Netherlands), respectively. Detailed information on the location of the monitoring site and the online instruments can be found elsewhere (Yu et al., 2021).

       To quantify the model performance, the Spearman's correlation coefficient (r) and the normalized mean bias (NMB) were calculated for 2-m temperature, wind speed, $SO_2$, $NO_2$, $O_3$ (ozone), and $PM_{2.5}$ at

each monitoring station listed in Table A1. The index of agreement (IoA) and the root mean square error (RMSE) were also calculated for the $PM_{2.5}$ component monitoring at the Pudong site.

### 3 Results and discussion

### 3.1 Changes in ship emissions under the staged fuel oil policies

       The emissions of $SO_2$, $NO_x$, CO, NMVOCs, $PM_{2.5}$, V, and Ni from shipping in the CECA and inland

waters of China from 2017 to 2021 were calculated based on the method introduced in Sect. 2.2.1 (Table




2). NO$_x$ is the major pollutant from shipping, and the nitrogen control policy has not been changed

nationwide in the study period. Hence, the amount of NO$_x$ emission can be regarded as a proxy of ship

traffic volume. As mentioned in Sect. 1, the variations of emissions in each April from 2017 to 2021 were

used to represent the interannual variations. In the CECA and inland waters of China, the NO$_x$ emissions

from shipping gradually increased from 2017 to 2020, with the largest increase (37.1%) from 2017 to

2018, and then slightly decreased in 2021. Due to the increase in ship activities, the NO$_x$ emissions from

shipping increased by 51.8% from 2017 to 2021. Figure S1 depicts the spatial distribution of NO$_x$

emissions from shipping in each April from 2017 to 2021 with high values along the major shipping

routes of coastal China, the Yangtze River and its main branches, and the Pearl River. In April 2019, the

higher emission intensity on the main route along the southeast coast of China was due to the bypass

behavior that ships tend to navigate outside the CECA. For the seasonal patterns, in 2017, the NO$_x$

emissions from shipping were higher in April (114.1 kt) and October (112.1 kt), smaller in July (101.6

kt) due to the fishing ban, and reached the lowest value in January (89.9 kt) which includes the Spring

Festival. However, in 2021, the lowest value (128.9 kt) occurred in July, while January exhibited

relatively high NO$_x$ emissions (169.2 kt) as the Spring Festival was in February.

By contrast, in the same area, the monthly average SO$_2$ (PM$_{2.5}$) emissions from shipping of January,

April, July, and October decreased by 68.4% (32.8%) from 2017 to 2021 due to the implementation of

the IMO Regulation and China's inland sulfur regulation. The monthly average PM$_{2.5}$ emissions reduced

from 7.6 kt in 2017 to 5.1 kt in 2021. In addition, the monthly average V emissions from shipping

experienced a dramatic drop (by 90.8%) from 118.8 t in 2017 to 43.9 t in 2021. The monthly average Ni

emissions decreased from 11.0 t in 2017 to 24.1 t in 2021, with a reduction of 42.0%. The average V/Ni

ratio decreased from 2.86 in 2017 to 0.46 in 2021. Figure S2 (Figure S3) shows the interannual variations

of V (Ni) emissions from shipping and anthropogenic sources excluding shipping. It can be clearly seen

that higher V and Ni emissions transferred from nearshore waters to the outer border of the CECA in

2019.

Table S7 shows the contributions of ship emissions to the total anthropogenic emissions in China's

200-nm zone and coastal provinces (coastal areas hereafter). The staged low sulfur policies since 2017

significantly reduced the SO$_2$, V, and Ni emissions, and their reduction rates were larger than those from

land-based anthropogenic sources, especially for V. The contributions of ship emissions to the total SO$_2$,

V, and Ni emissions in the coastal areas decreased from 13.9%, 89.2%, and 55.5% in 2017 to 7.7%,

 

56.0%, and 53.8% in 2021, respectively. The contribution of ship emissions to the total $PM_{2.5}$ emissions in the coastal areas remained at 4.0%. However, the share of $NO_x$ emissions from shipping increased from 13.2% in 2017 to 21.2% in 2021 due to the increase in shipping activities and the reduction in emissions from land-based sources.


**Table 2.** Time variation of emissions of $SO_2$, $NO_x$, CO, NMVOCs, $PM_{2.5}$, V, and Ni from shipping in the coastal emission control area (CECA) and inland waters of China.

|  | $SO_2$ (kt) | $NO_x$ (kt) | CO (kt) | NMVOCs (kt) | $PM_{2.5}$ (kt) | V (t) | Ni (t) |
|---|---|---|---|---|---|---|---|
| January 2017 | 35.3 | 89.9 | 4.2 | 4.6 | 6.5 | 102.1 | 35.7 |
| April 2017 | 44.4 | 114.1 | 5.4 | 6.0 | 8.3 | 130.7 | 45.7 |
| July 2017 | 40.3 | 101.6 | 4.6 | 5.1 | 7.2 | 115.0 | 40.2 |
| October 2017 | 43.5 | 112.1 | 5.3 | 5.9 | 8.2 | 127.4 | 44.7 |
| April 2018 | 63.2 | 156.4 | 7.3 | 8.1 | 11.5 | 185.9 | 64.7 |
| April 2019 | 13.2 | 166.7 | 8.1 | 8.9 | 5.3 | 51.1 | 29.1 |
| April 2020 | 14.3 | 170.9 | 8.1 | 9.0 | 5.7 | 15.3 | 32.0 |
| January 2021 | 13.8 | 169.2 | 8.2 | 8.6 | 5.4 | 12.1 | 26.4 |
| April 2021 | 13.9 | 164.8 | 8.0 | 8.3 | 5.4 | 11.8 | 25.9 |
| July 2021 | 10.4 | 128.9 | 6.1 | 6.2 | 4.0 | 8.2 | 18.3 |
| October 2021 | 13.6 | 171.0 | 8.4 | 8.9 | 5.5 | 11.7 | 25.7 |
| 2017 average per month | 40.9 | 104.4 | 4.9 | 5.4 | 7.6 | 118.8 | 41.6 |
| 2021 average per month | 12.9 | 158.5 | 7.7 | 8.0 | 5.1 | 11.0 | 24.1 |

Note: Average per month equals the average of emissions during January, April, July, and October.

### 3.2 Model performance

Uncertainties in simulation results of air quality can be caused by multiple factors such as the accuracy of meteorological inputs, uncertainties in emission inventories, and the simplification of mechanisms in the model. For meteorological elements, as shown in Table S8, the model can outstandingly reproduce the 2-m temperature in each city during the 11 simulated months from 2017 to 2021. Except for Qinzhou (r = 0.88), the r values of the other cities were all above 0.9, and over 0.95 for most cities. The

performance for relative humidity was slightly inferior to that of temperature, with correlations ranging from 0.7 to 0.8 in most cities. Except for Dalian (NMB = 2.4%), all other cities show negative biases, which was caused by the underestimation of nighttime radiation cooling. For the 10-m wind speed, overestimation was found in most cities especially megacities such as Shanghai (NMB = 95.5%) and Shenzhen (NMB = 72.2%). The WRF model with a resolution of 9 km can generally reproduce the

average wind direction at a station level which is significantly affected by local topography.

        For the concentration of pollutants, the simulation results from the base runs were used to evaluate the model performance of the CMAQ model. As shown in Table S9, large biases of over ±50% of the



concentration levels of $SO_2$ and $NO_2$ derived from the CMAQ were found in three megacities including Shanghai, Shenzhen, and Beijing with positive biases, and in Qinzhou with negative biases. These biases

were caused by the uncertainties in local emissions of the MEIC inventory. In general, the performance for the daily maximum 8-h average (MDA8) $O_3$ concentrations in the coastal cities showed the patterns of underestimation in Zhejiang province and to its north and overestimation in Fujian province and to its south, corresponding to the biases of $NO_2$ concentration. It is noted that the $PM_{2.5}$ concentrations were underestimated in all of the 21 port cities that we concerned, with an average NMB of -31.0% (Fig. 2a),

which was mainly attributed to the underestimation of secondary aerosols. For example, the NMB values of the concentrations of sulfate, nitrate, and ammonium in Shanghai reached -15.4%, -54.6%, and -21.8% respectively in the entire simulation periods (Table S10). Multiple causes can result in the underestimation of secondary aerosols especially nitrate such as lack of chemical mechanisms, underestimation of daytime $O_3$ and nighttime relative humidity, as well as overestimation of wind speed

(Sun et al., 2022; Xie et al., 2022) . The negative biases were larger in spring, which was likely due to insufficient consideration of heterogeneous reactions like reactions on the surface of mineral dust in the CMAQ model. Nevertheless, the model can reproduce the temporal variation of mean $PM_{2.5}$ concentrations, with a correlation of r = 0.80.

For the shipping-related $PM_{2.5}$ concentrations in the concerned port cities, the mean value during the

simulation periods was 1.6 μg m$^{-3}$ (Fig. 2b). The hourly mean shipping-related $PM_{2.5}$ peaked on the early morning of 27 July 2017 (local time, hereafter) with a concentration of 10.4 μg m$^{-3}$, while the minimum value of -0.9 μg m$^{-3}$ occurred on the morning of 2 January 2021. Shipping is an emission sector releasing large amounts of $NO_x$ which can participate in complex non-linear chemistry. The limitation of chemical mechanisms in the model could lead to systematic uncertainties. Ship-emitted $NO_x$ significantly

consumes atmospheric oxidants such as $O_3$ and various radicals (OH, $HO_2$, $RO_2$, etc.) in areas controlled by the VOC-limited regime. The potential reduction of these oxidants can inhibit the second aerosol formation, which is the major reason for the negative simulated values of $PM_{2.5}$ related to ship emissions.



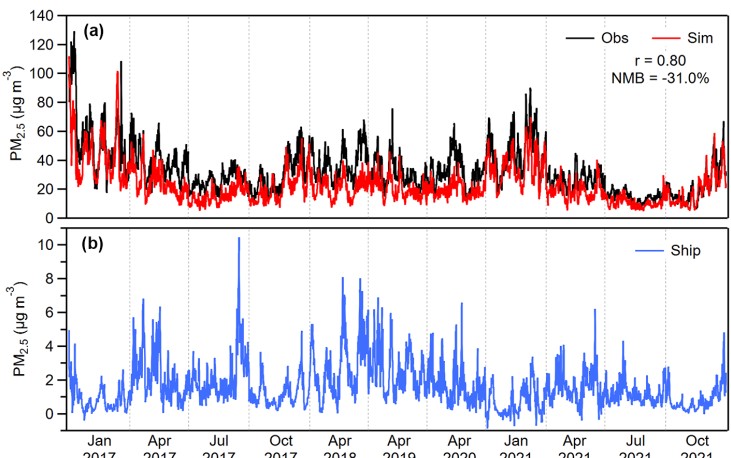

**Figure 2.** Time series of hourly mean PM$_{2.5}$ concentrations during the simulation periods, averaged for the representative port cities of China: **(a)** observational data (Obs) and simulated results of base runs (Sim), as well as **(b)** simulated shipping-related PM$_{2.5}$.

Regarding the tracers of ship emissions, as shown in Table S11, the simulated results of the monthly mean concentrations of V and Ni, as well as the Ni/V ratios are acceptable, thus the emission inventories of V and Ni established in this study being applicable. The model successfully reproduced the changing impacts of fuel oil policies on V and Ni. For the uncertainties, the model tended to overestimate the concentrations of the two metals in winter, such as overestimating the concentrations of V and Ni in January 2017 by 41% and 30%, respectively. The pattern in July 2021 was underestimating both V and Ni concentrations by 38%. These results can be brought by uncertainties in the simulation of diffusion conditions. The CMAQ model was fed by monthly ship emissions, and the high-frequency ship traffic data was smoothed, resulting the relatively weak temporal correlation between observational data and simulation results with hourly resolution. In addition, the uncertainties in meteorology can also affect the simulation of the transport process of ship emissions to urban areas. Nevertheless, the model can characterize the diurnal variation patterns of V (Fig. S4), with higher values during the nighttime and lower values during the daytime. The daily variations of simulation results were more pronounced than those based on observation, which was due to the overestimation of the diurnal cycle of the PBLH using the YSU PBL scheme (Du et al., 2020).

**3.3 Spatiotemporal patterns of the shipping-related gas precursors of PM$_{2.5}$**

SO$_2$ and NO$_2$ are the key gas precursors of secondary aerosols and the important pollutants from



shipping, and thus their spatiotemporal patterns are of interest. Figure 3 depicts the impacts of ship

emissions on the $SO_2$ concentration during the DECA 1.0 period (represented by 2017) and after the

implementation of the IMO Regulation (represented by 2021). In 2017, the spatial pattern of the $SO_2$

concentration from shipping performed high values along the main routes with the maximum value of

6.1 µg m$^{-3}$ (in a grid cell level) in the Yangtze River Estuary (YRE), and a decreasing trend toward both

sides (Fig. 3a). In 2021, the hotspots in China's coastal areas were located to the east of Shanghai and

Zhejiang, to the southeast of Fujian, and in the Pearl River Estuary (PRE), with the highest value of 2.5

µg m$^{-3}$ (Fig. 3b). With a remarkable decrease in fuel sulfur content (FSC) globally, the $SO_2$ concentration

from shipping reduced in all simulated areas (Fig. 3c). The largest reduction was observed in the lower

reaches of the Yangtze River reaching 5.2 µg m$^{-3}$ due to the fuel type shift to the ULSFO.

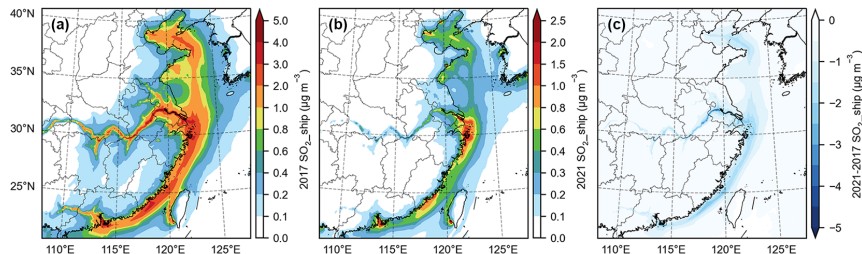


**Figure 3.** Impacts of ship emissions on the $SO_2$ concentrations (SO2_ship) in **(a)** 2017 and **(b)** 2021, as well as its **(c)** absolute change from 2017 to 2021. The annual average equals the average of January, April, July, and October here and hereafter.

As $SO_2$ is a typical primary pollutant emitted by ships, it was selected to analyze the factors affecting

the seasonal and interannual patterns such as emission intensity and meteorological conditions. With

respect to the seasonal patterns, the highest $SO_2$ concentrations from shipping in most regions of China

occurred in spring when the prevailing wind was weak and generally blew onshore in the areas with high

ship emissions (Fig. S5). Meanwhile, the ship emissions in spring were in a high level with ~8% higher

than the annual average. Hence, we selected April as the representative month for the analysis of

interannual variations. In winter, the concentrations on the land showed higher values due to the weak

diffusion conditions. In the offshore areas, the concentrations in winter were not high as those in spring,

which was caused by the significant winter monsoon. In summer, despite the better diffusion conditions,

the concentrations in the areas with dense ship emissions were only second to those in spring in 2017.

However, the concentrations shared relatively low levels in summer and autumn in 2021. This difference




can be attributed to the direction and the intensity of the summer monsoon. In the summer of 2021, the

strong southeasterly wind was in favor of the diffusion of ship emissions. Although the ship emissions in

autumn were comparable to those in spring, the concentrations in autumn were much lower than those in

spring in marine areas, which was resulted from the northeasterly prevailing wind diluting the pollutants

emitted by ships.

Figure S6 shows the variations of the $SO_2$ concentrations from shipping in each April from 2017 to

2021. The overall concentrations experienced staged reduction with the changes of the fuel oil policies.

However, due to large variations in local emissions, the maximum value in each year presented a

fluctuating trend with 7.3, 30.1, 7.1, 11.4, and 4.3 μg m$^{-3}$ from 2017 to 2021. The hotspots located in the

central part of the coast of Zhejiang in 2018 and 2020 were likely due to the intensive fishing activities.

Besides, the simulated wind field patterns in the coastal areas were similar in April of 2017, 2018, 2019,

and 2021. However, in April 2020, a stable high-pressure system over the Yellow Sea led to a better

horizontal diffusion condition for the emissions on the main routes.

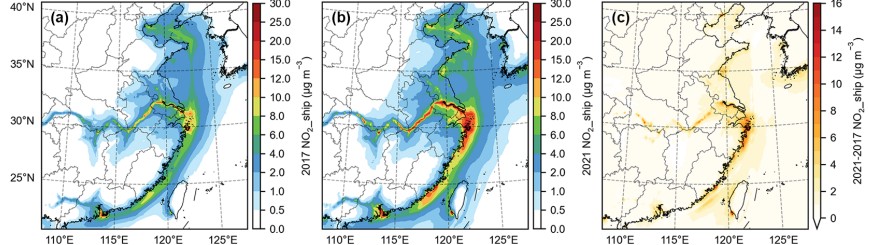

**Figure 4.** Impacts of ship emissions on the NO$_2$ concentrations (NO$_2$_ship) in **(a)** 2017 and **(b)** 2021, as well as its
**(c)** absolute change from 2017 to 2021.

The main source of the ambient NO$_2$ from shipping is the rapid oxidation of NO in ship plumes, and

the high concentrations still fixed on the main routes and ports in the lower reaches of the Yangtze River

and the PRE, as well as coastal Shanghai, Zhejiang, and Fujian (Fig. 4). The maximum values in 2017

and 2021 were 17.9 and 26.0 μg m$^{-3}$, respectively. The concentrations increased in most areas from 2017

to 2021, and the increase was larger near the ports of Ningbo-Zhoushan in Zhejiang, Quanzhou in Fujian,

Gaoxiong in Taiwan, and Tangshan in Hebei with the maximum of 15.4 μg m$^{-3}$. The decrease in SO$_2$ and

the increase in NO$_2$ were both significant from 2017 to 2021, which is expected to affect the formation

pathways of shipping-related secondary aerosols and the composition of shipping-related PM$_{2.5}$.



### 3.4 Spatiotemporal patterns of shipping-related PM$_{2.5}$ and its components

#### 3.4.1 Fine particulate matter (PM$_{2.5}$)

PM$_{2.5}$ related to ship emissions contains primary and secondary aerosols, and the secondary aerosols lead to the difference in the spatial patterns between the gas precursors and PM$_{2.5}$. As shown in Fig. 5a, the concentrations of shipping-related PM$_{2.5}$ in 2017 displayed values of over 2 μg m$^{-3}$ along the main routes from the Yellow Sea to coastal Fujian, with the maximum of 3.8 μg m$^{-3}$ near Zhoushan Islands. The areas with concentration over 1 μg m$^{-3}$ covered most of the Yellow and Bohai Seas and extended to the Middle Yangtze River (~1000 km away from the coast). However, the areas with concentration over 1 μg m$^{-3}$ shrunk to the coast and the central Yangtze River Basin in 2021, with the maximum of 2.6 μg m$^{-3}$ in the eastern Shandong Peninsula (Fig. 5b). The areas with shipping-related PM$_{2.5}$ of over 0.1 μg m$^{-3}$ occupied most of the model domain both in 2017 and 2021, which was markedly different from the patterns of SO$_2$ and NO$_2$ from shipping. This result is caused by sustained aging processes during the transport of ship-emitted pollutants as well as the longer lifetime of PM compared to SO$_2$ and NO$_2$ (Seinfeld and Pandis, 2016). The decrease in the concentration of shipping-related PM$_{2.5}$ showed the largest value of 1.9 μg m$^{-3}$ in the sea area of Zhoushan Islands, while the slight increase within 0.5 μg m$^{-3}$ was found in very small areas in Hunan, Shandong, and Liaoning provinces (Fig. 5c). The result in Lv et al. (2018) showed that the increased PM$_{2.5}$ concentration in China caused by shipping was up to 5.2 μg m$^{-3}$ in 2015. The values in this study were 3.8 μg m$^{-3}$ in 2017 and 2.6 μg m$^{-3}$ in 2021, demonstrating the decreasing trend of the shipping-related PM$_{2.5}$ concentration under the staged fuel oil policies in China.

For the relative potential impact of ship emissions on PM$_{2.5}$, also called the contribution of ship emissions to ambient PM$_{2.5}$, in 2017, larger values of over 10% distributed along the main routes from the southern Yellow Sea to the northern South China Sea, with the maximum of 21.3% in the Taiwan Strait (Fig. 5d). The distance between the land areas with the relative impact of over 4% and the coast could be up to ~300 km. The relative impact was not less than 1% in the model domain except for northwestern China. Lower values in remote marine areas were related to the contribution of sea salt. In 2021, the relative impact values exceeding 10% were found only along China's southeast coast and in the eastern end of Shandong Peninsula, with the maximum of 16.9% (Fig. 5e). In the areas with high relative impact values mentioned above, the relative impact decreased remarkably, with the largest




decrease of -11.0 percentage points (Fig. 5f). However, an increase in the relative impact was observed

near the south and east coast of Shandong with the maximum of 3.4%, and small positive changes

scattered in China's inland areas. This result may be caused by the increase in nitrate formation related

to shipping and the decrease in land-based anthropogenic emissions, which will be discussed in Sect.

3.4.3.

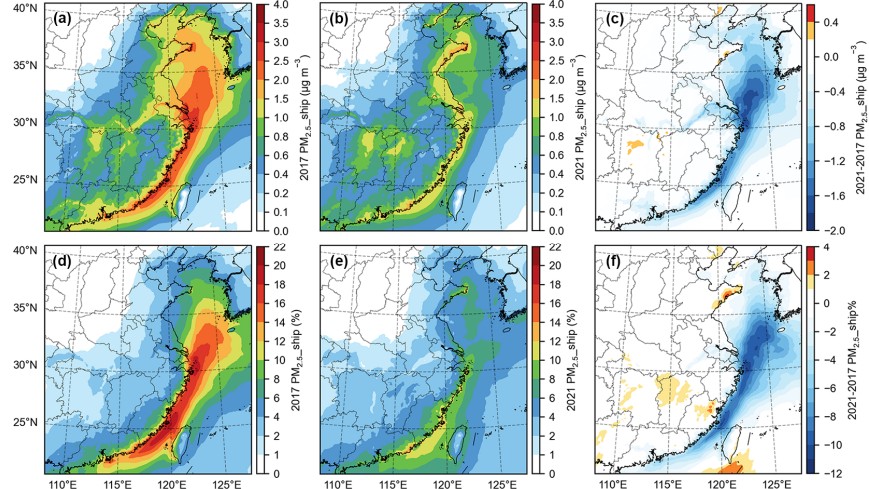


**Figure 5.** Potential impacts of ship emissions on PM$_{2.5}$ (PM$_{2.5}$_ship): for concentration (in μg m$^{-3}$) in **(a)** 2017 and **(b)** 2021, as well as **(c)** the absolute change from 2017 to 2021; and for contribution (in %) in **(d)** 2017 and **(e)** 2021, as well as **(f)** the change of percentage from 2017 to 2021 (in percentage point).

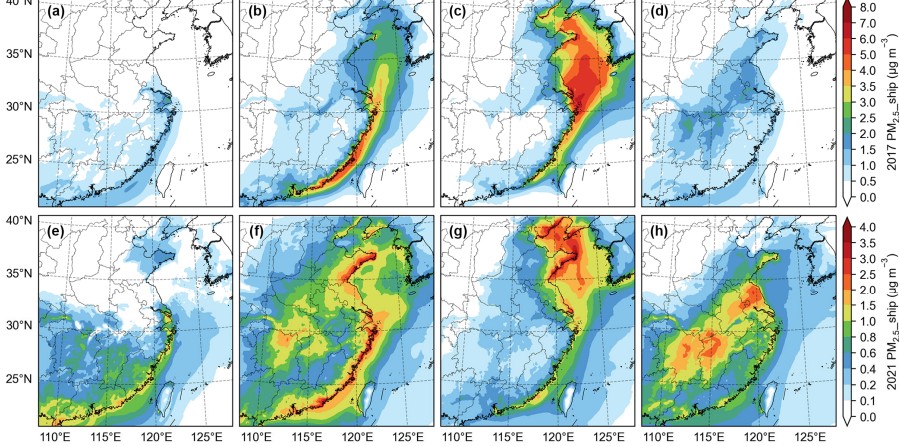

**Figure 6.** Seasonal patterns of the potential impacts of ship emissions on PM$_{2.5}$ (PM$_{2.5}$_ship) concentrations in **(a)** January, **(b)** April, **(c)** July, and **(d)** October of 2017; and **(e)** January, **(f)** April, **(g)** July, and **(h)** October of 2021.





The seasonal patterns of the concentrations of shipping-related $PM_{2.5}$ in 2017 and 2021 were rather different from those of the $SO_2$ and $NO_2$ concentrations from shipping (Fig. 6). During the springtime, relatively high shipping-related $PM_{2.5}$ concentration was located along the southeast coast both in 2017

and 2021 as well as the coast of Shandong in 2021. The pollutants emitted by ships were prone to accumulate due to the meteorology and experience chemical reactions in coastal areas in spring. Unexpectedly, the hotspots transferred to sea areas in summer, and the maximum values were even higher than those in spring for both years. As the diffusion conditions are better in summer, there should be other factors like secondary organic aerosol (SOA) formation (see Sect. 3.4.4).

In autumn, cold air masses from northeastern Asia diluted the gas precursors in coastal areas. The atmosphere over inland areas were in an ammonia-rich condition favoring nitrate formation, and thus relatively high values shifted to inland areas. During the wintertime, the shipping-related $PM_{2.5}$ concentration showed moderate levels in Shanghai both in 2017 and 2021, as well as along the southeastern to southern coast in 2021. It is worth noting that the values in most northern China did not

exceed 0.5 $\mu g\ m^{-3}$ and even lower than 0.1 $\mu g\ m^{-3}$ in some coastal areas, which was very different from the pattern of $SO_2$ from shipping. Weaker diffusion conditions and lower temperature as well as higher nocturnal humidity led to higher efficiency of secondary aerosol formation. However, $NH_3$ is sufficiently consumed by $SO_2$ and $NO_x$ from land-based emissions, and thus the formation of secondary aerosols related to shipping is inhibited, which called the competitive mechanism by land-based sources. As

discussed in Sect. 3.3, meteorology is a much more important factor affecting the seasonal patterns of the $SO_2$ and $NO_2$ concentrations from shipping compared to emissions. However, the conversion rate of gas precursors is a key factor affecting the seasonal pattern of the concentration of shipping-related $PM_{2.5}$, while the influence of diffusion conditions is relatively small.

For the relative impact of shipping-related $PM_{2.5}$, the seasonal pattern generally showed a decreasing

trend in summer, spring, autumn, and winter in the model domain, which was mainly due to the impact of the East Asian monsoon on the relative spatial distribution of ship and land-based emissions (Fig. 7). Despite the fact that the concentration of shipping-related $PM_{2.5}$ in coastal areas peaked in spring, the highest relative impact was recorded in summer. In the summer of 2017, the largest relative impact exceeded 50% and 25% over the marine and land areas of the southeastern coastal areas, respectively. In

the summer of 2021, the areas with higher absolute and relative impacts of shipping-related $PM_{2.5}$ shifted northward to the Yellow Sea, which was driven by the southeasterly summer monsoon. Among the





concerned port cities, in the summer of 2017, Zhoushan showed the highest value of 37.6%, while the lowest value of 4.8% for Qinzhou. In the summer of 2021, the highest (25.0%) and the lowest values (4.2%) occurred in Zhoushan and Nanjing, respectively. In comparison, the relative impact was much

smaller in winter, with the highest value of 5.7% in Ningbo and the lowest value of 0.14% in Caofeidian in the winter of 2017, and the highest value of 6.2% in Fuzhou and the lowest value of 0.04% in Nanjing in the winter of 2021.

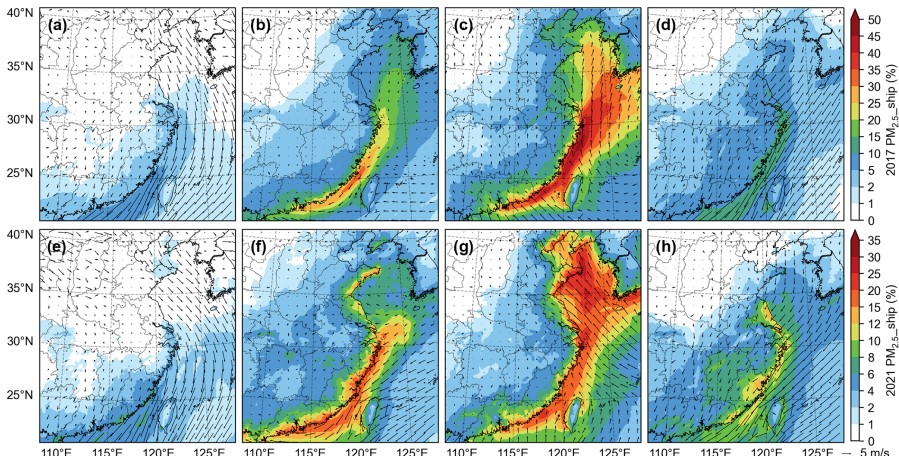

**Figure 7.** Seasonal patterns of the potential impacts of ship emissions on $PM_{2.5}$ ($PM_{2.5}$_ship) for contribution as well as the simulated monthly average wind fields in **(a)** January, **(b)** April, **(c)** July, and **(d)** October of 2017; and **(e)** January, **(f)** April, **(g)** July, and **(h)** October of 2021.

Figure 8 shows the spatial patterns of the concentration of shipping-related $PM_{2.5}$ in the spring of 2017−2021. In April 2017, the highest concentration of 7.2 μg m$^{-3}$ was located over the coastal waters of

Fujian. In April 2018, the highest concentration rose to 10.4 μg m$^{-3}$; the concentration generally increased in the model domain, especially over the PRE and the coastal waters of Zhejiang, which was due to the increase in shipping activities. In April 2019, high values were closer to the coastline compared to the DECA 1.0 period, and the CECA border was indistinct compared to the pattern of the $SO_2$ concentration from shipping. This result highlights the crucial role of secondary aerosols in the shipping-related $PM_{2.5}$

in the presence of plenty $NH_3$ emissions. The highest concentration reduced to 6.5 μg m$^{-3}$, whereas there was an increase along the coast of Shandong due to more secondary aerosol formation. In April 2020, due to the implementation of the IMO Regulation, the concentration declined in most marine areas especially the Yellow Sea in which the main routes are not included in the CECA before 2020. However,





the concentration increased along coastal Zhejiang, and the highest value in the model domain rebounced

to 8.3 μg m$^{-3}$, which was in accord with the patterns of the SO$_2$ and NO$_2$ concentrations from shipping.

This result was likely due to the recovery of fishing activities after the COVID-19 lockdown. In April

2021, ship emissions were more evenly distributed, and hence the concentration of shipping-related PM$_{2.5}$

only presented relatively high values in a city level, with the maximum of 4.5 μg m$^{-3}$ in coastal Zhejiang.

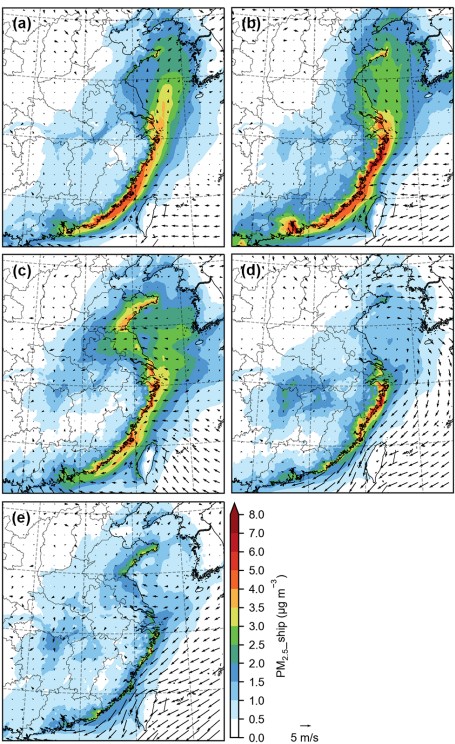


**Figure 8.** Interannual variations of the potential impacts of ship emissions on PM$_{2.5}$ (PM$_{2.5}$_ship) concentrations as well as the simulated monthly average wind fields in April of **(a)** 2017, **(b)** 2018, **(c)** 2019, **(d)** 2020, and **(e)** 2021.

### 3.4.2 Trace elements (V and Ni)

V and Ni are strongly correlated with SO$_2$ among the chemical species emitted by ships, and thus their

concentrations from shipping share similar spatial patterns. In the CMAQ model, V and Ni do not

participate in any chemical reactions, while SO$_2$ can be oxidized to sulfuric acid (H$_2$SO$_4$) and sulfate.

Therefore, emissions and meteorology are the factors affecting the concentrations of PM$_{2.5}$-bound V and

Ni from shipping. In 2017, the areas with the V concentration from shipping exceeding 0.5 ng m$^{-3}$ covered

most parts of eastern China, with the maximum of 24.5 ng m$^{-3}$ (Fig. 9a). In contrast, for the land areas,



only coastal areas and the Yangtze River showed values over 0.1 ng m$^{-3}$ in 2021, and the maximum along

the coast substantially decreased to 2.2 ng m$^{-3}$ (Fig. 9b). Ship emissions overwhelmingly dominated the

V concentration in the marine and coastal areas as well as along the Yangtze River in 2017 (Fig. 9c).

However, in 2021, the contribution of land-based emissions was closed to or even higher than that of

ship emissions in the coastal areas (Fig. 9d). There was also an evident decrease in the contribution of

ship emissions in remote marine areas because V from land-based emissions including anthropogenic

sources and mineral dust can be transported by the high-altitude westerly wind.

The spatial pattern of the Ni concentration from shipping was similar to that of the V concentration

from shipping, while their relative changes from 2017 to 2021 were different (Fig. 10). The highest Ni

concentration from shipping was 7.8 ng m$^{-3}$ in 2017 and decreased to 4.1 ng m$^{-3}$ in 2021. The V/Ni ratios

in ambient particles from shipping decreased from ~3.0 to ~0.5 from 2017 to 2021. In 2017, the

contribution of ship emissions to the Ni concentration was lower compared to V. Over land areas, the

contour of 80% for the V concentration share generally corresponded to the contour of 40% for the Ni

concentration share. It is noted that the reduction in the Ni concentration share from 2017 to 2021 was

small in the model domain, and the Ni concentration share overtook the V concentration share. Despite

the sharp reduction in the concentrations of V and Ni from shipping, shipping is still an important source

of the ambient V and Ni under the current fuel oil regulations.

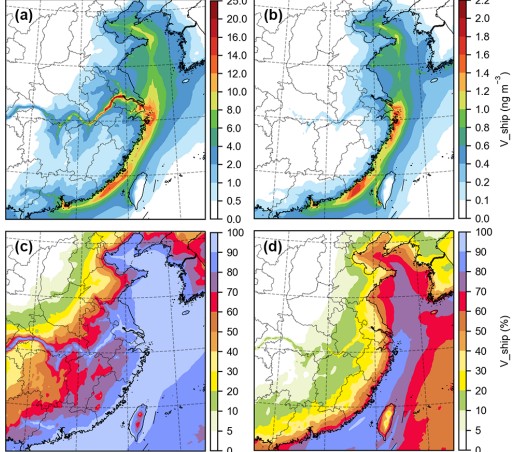

**Figure 9.** Impacts of ship emissions on V (V_ship): for concentration (in ng m$^{-3}$) in PM$_{2.5}$ in **(a)** 2017 and **(b)** 2021;
and for contribution (in %) in **(c)** 2017 and **(d)** 2021.



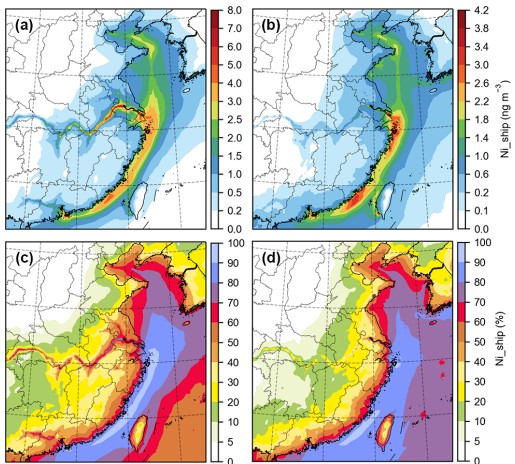

**Figure 10.** Impacts of ship emissions on Ni (Ni_ship): for concentration (in ng m$^{-3}$) in PM$_{2.5}$ in **(a)** 2017 and **(b)** 2021; and for contribution (in %) in **(c)** 2017 and **(d)** 2021.

### 3.4.3 Secondary inorganic aerosols

Sulfate (SO$_4^{2-}$), nitrate (NO$_3^-$), and ammonium (NH$_4^+$), known as SNA for short, are the most important secondary inorganic species in PM$_{2.5}$. Their concentrations are highly related to the concentrations and the conversion rates of the gas precursors including SO$_2$, NO$_x$, and NH$_3$. For the formation of sulfate, the main atmospheric oxidant is hydroxyl radical (OH) which transforms SO$_2$ to H$_2$SO$_4$. Then, H$_2$SO$_4$ is neutralized by NH$_3$ forming (NH$_4$)$_2$SO$_4$, or dissolves in aerosol liquid water and binds with positive ions like Na$^+$, leading to the higher shipping-related SO$_4^{2-}$ concentration from the southern Yellow Sea to coastal Fujian in 2017 (Fig. 11a). The maximum of 1.2 μg m$^{-3}$ was located in the waters of Zhoushan Islands. Taking Zhoushan as an example, we found that primary sulfate emitted by ships played a minor role and secondary sulfate accounted for 91.8% of the shipping-related sulfate. As EC is a stable species, the primary sulfate concentration from shipping can be calculated through multiplying the EC concentration from shipping by the ratio of SO$_4^{2-}$ to EC in ship-emitted PM in Table S3. Thus, the secondary part of shipping-related sulfate equals the difference of the total concentration and the primary part. Although the SO$_2$ concentration from shipping was also at a relatively high level along the Yangtze River, the shipping-related SO$_4^{2-}$ concentration was significantly lower compared to the values in sea areas, which was attributed to the competitive mechanism by land-based emissions. In 2021, the shipping-related SO$_4^{2-}$ concentration decreased with the reduction in SO$_2$ emissions, with the maximum of 0.46 μg m$^{-3}$. There were unexpected negative values over land areas, which was likely due to the



depletion of oxidants to generate nitric acid (HNO$_3$) in areas characterized by high-NO$_x$ conditions.

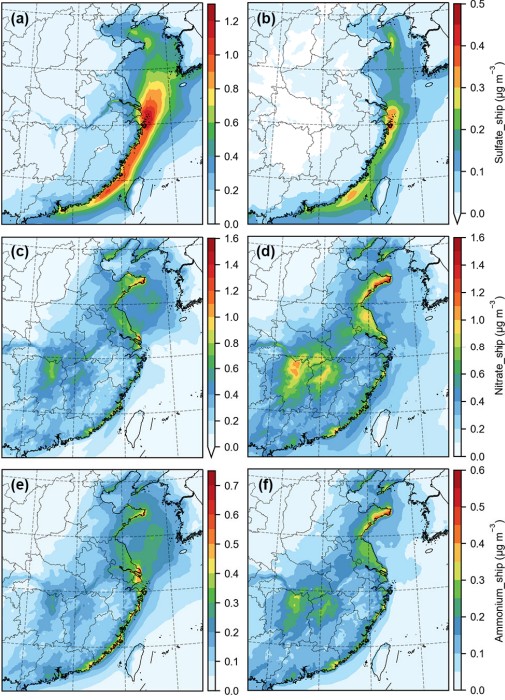

**Figure 11.** Potential impacts of ship emissions on sulfate (Sulfate_ship), nitrate (Nitrate_ship), and ammonium (Ammonium_ship): for sulfate in **(a)** 2017 and **(b)** 2021; for nitrate in **(c)** 2017 and **(d)** 2021; and for ammonium in **(e)** 2017 and **(f)** 2021.

Most of initially emitted NO$_x$ is in the form of NO, and is rapidly oxidized to NO$_2$. During the daytime, NO$_2$ can be converted to HNO$_3$ by reacting with OH radicals. During the nighttime, NO$_2$ combines with NO$_3$ radicals originating from the oxidization of NO$_2$ by O$_3$ to generate dinitrogen pentoxide (N$_2$O$_5$). HNO$_3$ is formed via the reaction of N$_2$O$_5$ and water. Compared to HNO$_3$, NH$_3$ preferentially reacts with H$_2$SO$_4$ because (NH$_4$)$_2$SO$_4$ is more stable than NH$_4$NO$_3$. Therefore, the formation of particulate nitrate requires sufficient amount of NH$_3$. The differences in the formation mechanisms between sulfate and nitrate can explain the significant differences in the spatial patterns of the concentrations between SO$_4^{2-}$ and NO$_3^-$ related to ship emissions. The shipping-related NH$_4$NO$_3$ in the marine atmosphere came from NH$_3$ transported from land areas or the transport of NH$_4$NO$_3$ already formed in coastal areas. In marine areas, the concentration of shipping-related NO$_3^-$ only displayed moderate levels over the Yellow Sea in 2017. In comparison, higher concentration levels were concentrated along the coast, with the maximum



of 1.4 μg m$^{-3}$ (Fig. 11c). In 2021, the values increased over land areas with the increase in NO$_x$ emissions

and the decrease in SO$_2$ emissions from shipping, while the maximum slightly increased to 1.5 μg m$^{-3}$

located along the coast of Shandong (Fig .11d). There was a noticeable increase in the central Yangtze

River Basin, corresponding to the negative values of shipping-related SO$_4^{2-}$. In this study, there was no

marked negative value of shipping-related NO$_3^-$ in the model domain, which differed from the result in

the Mediterranean Sea also based on the CMAQ model (Fink et al., 2023a). This was caused by the huge

amounts of NH$_3$ emissions from agriculture in China, providing an ammonia-rich condition to generate

NH$_4$NO$_3$.

The spatial patterns of shipping-related NH$_4^+$ were similar with those of shipping-related NO$_3^-$ both in

2017 and 2021 (Fig. 11e and Fig. 11f). This result implied that the relatively high levels of shipping-

related SO$_4^{2-}$ in ammonia-poor offshore areas tended to present in the form of metal salt like Na$_2$SO$_4$

rather than (NH$_4$)$_2$SO$_4$. Compared to SO$_4^{2-}$ and NO$_3^-$, the concentration of shipping-related NH$_4^+$ was at

a lower level due to the much smaller molar mass, with the maxima of 0.73 μg m$^{-3}$ in 2017 and 0.58 μg

m$^{-3}$ in 2021. Overall, higher levels of potential impacts of ship emissions on SO$_4^{2-}$ occurred in marine

areas, while in land areas for NO$_3^-$ and NH$_4^+$. Given that PM$_{2.5}$ especially secondary species like nitrate

was underestimated in China's coastal cities in this study as discussed in Sect. 3.2, the simulated

concentrations of SNA related to ship emissions were conservative.

### 3.4.4 Organic aerosols

Organic aerosol (OA) is categorized into primary organic aerosol (POA) and secondary organic aerosol

(SOA). POA is calculated as the sum of particulate organic carbon (POC) and particulate non-carbon

organic matter (PNCOM), two species in the AERO7 module. POA and EC share similar atmospheric

processes in the CMAQ model. In this study, the sum of the contributions of POC and PNCOM to primary

PM$_{2.5}$ emitted by ships using HSFO was set to 70.8%, while this value was reduced to 33.4% in the case

of burning LSFO (Table S3). Thus, the reduction in the POA concentration from shipping due to the fuel

type change was more significant than those in the primary PM$_{2.5}$ concentration from shipping and also

the total shipping-related PM$_{2.5}$ concentration. The maximum value decreased from 0.41 μg m$^{-3}$ in 2017

to 0.12 μg m$^{-3}$ in 2021 (Fig. S7). The contribution of ship emissions to the ambient POA concentration

showed higher values compare to the PM$_{2.5}$ concentration share, with the maximum of 41.9% in 2017

and 29.4% in 2021. The transport of secondary aerosols related to land-based emissions as well as the





restricted secondary aerosol formation in marine areas with the NH$_3$-poor conditions led to lower values

of the PM$_{2.5}$ concentration share.

SOA, a kind of photochemical product, is produced via the reactions of VOCs or semi-volatile organic compounds (SVOCs) with atmospheric oxidants. As discussed in Sect. 3.2, this study showed the potential impacts of ship emissions on atmospheric photochemistry by using the zero-out method, which differed from the situations of the real-world atmosphere such as the chemical processes in a single ship

plume. The potential impacts of ship emissions on the SOA and SO$_4^{2-}$ concentrations shared similar spatial patterns in 2017, with the maximum of 1.3 μg m$^{-3}$ over the waters of Zhoushan Islands (Fig. 12a). No negative value was found in the entire model domain. However, in 2021, the marine and coastal areas displayed positive values with the maximum of 0.33 μg m$^{-3}$ near the eastern coast of the Pearl River Delta, whereas negative values were obtained in the central Yangtze River Basin with the minimum of -

0.33 μg m$^{-3}$ (Fig. 12b).

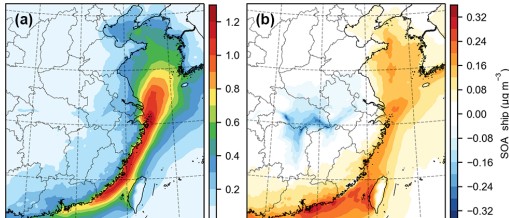

**Figure 12.** Potential impacts of ship emissions on secondary organic aerosols (SOA_ship) in **(a)** 2017 and **(b)** 2021.

For the seasonal patterns of the shipping-related SOA concentration shown in Fig. S8, the highest level

in the eastern offshore areas of China was found in the summer of both 2017 and 2021. In summer, high temperature and good lighting conditions are in favor of BVOC emissions and photochemical reactions, resulting in abundant oxidants in the background atmosphere to generate O$_3$ and SOA. Ship emissions could increase the concentrations of O$_3$ and SOA in the NO$_x$-limited marine atmosphere with low NO$_x$ concentration. Although the NO$_x$ and VOCs emissions from shipping increased from 2017 to 2021, the

shipping-related SOA concentration during the summertime decreased. This result suggested that gas-particle partitioning of organic matter with low to medium volatility played an important role in SOA formation. The significant decrease in the shipping-related PM was likely to reduce the impact of gas-particle partitioning. In contrast to the pattern in summer, the potential impact of ship emissions on SOA showed negative values in inland areas in the winter of 2021, which was indirectly caused by the O$_3$





titration by $NO_x$ under the VOC-limited regime. The potential impact was insignificant in the other

seasons, and thus the annual mean values in central China were negative. The increasing $NO_x$ emissions

along the Yangtze River led to more depletion in oxidants over inland areas during the wintertime. The

minimum shipping-related SOA concentration in winter decreased from -0.45 µg m$^{-3}$ in 2017 to -1.0 µg

m$^{-3}$ in 2021.

**3.5 Intercomparison of the impacts over the port cities**

    In Sect. 3.5, the perspective will be shifted from regional scale to urban scale. For each of the 21 port

cities selected in this study, the data of the grids where populous downtown areas (instead of the ports

themselves) are located was extracted for the analysis on the impacts of ship emissions. Thus, the results

can reflect the role of ship emissions in the presence of large amounts of land-based emissions in urban

areas.

    **3.5.1 Effects of the implementation of the IMO Regulation**

    Figure 13 shows the absolute and relative impacts of ship emissions on multiple species such as $SO_2$,

$NO_2$, $PM_{2.5}$, $SO_4^{2-}$, $NO_3^-$, $NH_4^+$, carbonaceous aerosol (CA, referred to the sum of OA and EC), V, and Ni

at the annual average level in China's main port cities in 2017 and 2021. On average across all the

concerned cities, the $SO_2$ concentration from shipping reduced from 1.3 µg m$^{-3}$ in 2017 to 0.48 µg m$^{-3}$ in

2021 due to the implementation of the IMO Regulation. Its reduction rate was 63.3%, closed to the

reduction rate of $SO_2$ emissions from shipping in the CECA and inland areas of China (68.4%). The share

of the $SO_2$ concentration from shipping was 14.3% in 2017 and decreased to 9.0% in 2021. However, the

$NO_2$ concentration from shipping increased from 4.3 µg m$^{-3}$ to 7.1 µg m$^{-3}$ due to the growth in shipping

activities. The minima of the $SO_2$ and $NO_2$ concentrations from shipping were both observed in Qinzhou.

The increase rate (65.9%) was higher than that of $NO_x$ emissions from shipping (51.8%), because ship

emissions increased the atmospheric oxidation capacity in offshore areas and more NO could be oxidized

to $NO_2$. The share of the $NO_2$ concentration from shipping rose from 13.9% to 22.6%.

    The shipping-related $PM_{2.5}$ concentration reduced from 1.6 µg m$^{-3}$ to 1.1 µg m$^{-3}$, and the reduction rate

(32.7%) was very closed to that of $PM_{2.5}$ emissions from shipping (32.8%). The shipping-related $PM_{2.5}$

concentrations in the city level ranged from 0.84 µg m$^{-3}$ in Qinzhou to 2.7 µg m$^{-3}$ in Zhoushan in 2017,

while from 0.63 µg m$^{-3}$ in Yingkou to 1.6 µg m$^{-3}$ in Qingdao in 2021. It is noted that only Qinzhou



experienced an increase, though very small (0.04 µg m⁻³), which was corresponding to the intense growth in cargo throughput of the Beibu Gulf ports approaching 100% in the past six years. After the operation

of the Pinglu Canal at the end of 2026, there is still great potential for the growth in shipping activities and thus the impacts of ship emissions on the air quality in Qinzhou will be much more significant if the current policies do not change. Compared to $SO_2$ and $NO_2$, the contribution of ship emissions to the total $PM_{2.5}$ concentration only showed a slight change from 6.8% to 5.5%. The shipping-related $PM_{2.5}$ shares in the city level ranged from 3.0% in Yingkou to 17.4% in Zhoushan in 2017, while from 2.5% in Nanjing

to 10.3% in Zhoushan in 2021. Unexpectedly, they showed slight increases in five cities adjacent to the Yellow Sea including Dalian, Yantai, Qingdao, Rizhao, and Lianyungang, which was in accord with the increase in the shipping-related $NO_3^-$ and $NH_4^+$ concentration shares.

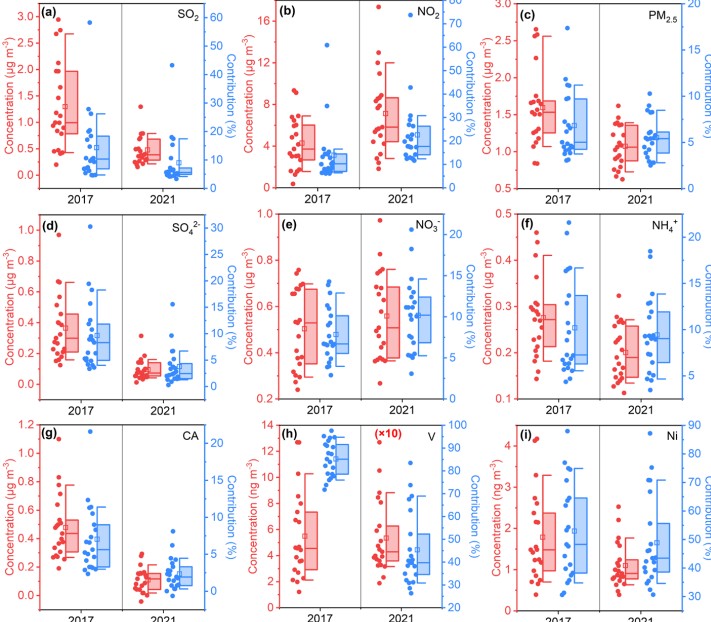

**Figure 13.** Impacts of ship emissions on **(a)** $SO_2$, **(b)** $NO_2$, **(c)** $PM_{2.5}$, **(d)** $SO_4^{2-}$, **(e)** $NO_3^-$, **(f)** $NH_4^+$, **(g)** carbonaceous aerosol (CA), **(h)** V, and **(i)** Ni over the representative port cities of China in 2017 and 2021. Left axis and right axis show concentration and contribution, respectively. Box plots show the mean (square), median (-), lower and upper quartile (boxes), and the 10th and the 90th percentiles (whiskers) of the simulated results. The V concentration in 2021 is multiplied by 10.

For the chemical species in the shipping-related $PM_{2.5}$, the $SO_4^{2-}$ concentration reduced from 0.36 µg m⁻³ to 0.09 µg m⁻³, with the reduction rate of 74.0%, higher than that of the $SO_2$ concentration from



shipping. In contrast, the shipping-related $NO_3^-$ concentration increased by 11.0% from 0.50 µg m$^{-3}$ to 0.56 µg m$^{-3}$. Nevertheless, its increase rate was much smaller than that of the $NO_2$ concentration from shipping, indicating a low nitrogen oxidation rate (NOR). Besides the increase in the $NO_x$ emissions from

shipping, the decrease in $SO_2$ provided more opportunities for the neutralization of $HNO_3$ by $NH_3$ to form $NH_4NO_3$. The weakened sulfate formation was demonstrated by the decrease in the contribution of the secondary $SO_4^{2-}$ to the total $SO_4^{2-}$ related to shipping from 89.8% to 83.9%. Because the decrease in $SO_4^{2-}$ was more significant than the increase in $NO_3^-$, the shipping-related $NH_4^+$ concentration decreased by 27.5% from 0.28 µg m$^{-3}$ to 0.20 µg m$^{-3}$. The shipping-related CA concentration decreased from 0.48

µg m$^{-3}$ to 0.11 µg m$^{-3}$, with a reduction rate of 76.9%, even higher than that of $SO_4^{2-}$. The V concentration from shipping sharply dropped from 5.5 ng m$^{-3}$ to 0.53 ng m$^{-3}$. In comparison, the Ni concentration from shipping decreased moderately from 1.8 ng m$^{-3}$ to 1.1 ng m$^{-3}$. The reduction rates of the V and Ni concentrations from shipping were 90.3% and 38.4% respectively, closed to the reduction rates of the emissions of V (90.8%) and Ni (42.0%) from shipping. In addition, the relative impacts of ship emissions

on the concentrations of $SO_4^{2-}$, $NO_3^-$, $NH_4^+$, CA, V, and Ni were 9.6%, 7.9%, 10.2%, 7.0%, 85.2%, and 52.9% in 2017 and 3.8%, 10.2%, 9.4%, 2.4%, 45.4%, and 48.9% in 2021, respectively. Among the six species, only $NO_3^-$ exhibited an increasing trend in the relative impact.

     Figure 14 presents the simulated chemical speciation of $PM_{2.5}$ from all sectors and shipping over China's main port cities in 2017 and 2021. Six categories of chemical species contributing more than 95%

to $PM_{2.5}$ were considered which include $SO_4^{2-}$, $NO_3^-$, $NH_4^+$, POA, SOA, and Soil. Soil, a variable in the CMAQ model output, is calculated following Eq. (1):

$$Soil = 2.20 \times Al + 3.48 \times Si + 1.63 \times Ca + 2.42 \times Fe + 1.94 \times Ti \qquad (1)$$

     The change in the characteristics of the shipping related $PM_{2.5}$ components was far more significant compared to the $PM_{2.5}$ derived by the base runs. $SO_4^{2-}$ accounted for 21.0% in the shipping-related $PM_{2.5}$

averaged for the concerned port cities in 2017 and decreased to 8.9% in 2021, while the percentage of $SO_4^{2-}$ in the $PM_{2.5}$ from all sectors only decreased from 16.1% to 14.7%. This result is regarded as the potential impact and does not suggest that ship-emitted $PM_{2.5}$ contains less sulfate content than $PM_{2.5}$ from land-based sources. In the low-sulfur era, ship emissions tend to enhance the nitrate formation in the high-$HNO_3$ and high-$NH_3$ but low-$H_2SO_4$ conditions in China's coastal cities. Accordingly, $NO_3^-$ has

become the major component of the shipping-related $PM_{2.5}$, with the average $NO_3^-$ share increased from 30.7% to 54.6%. The $NH_4^+$ share also showed an increasing trend from 17.0% to 19.6% as the decrease



in $SO_4^{2-}$ cannot offset the increase in $NO_3^-$. In contrast, in the $PM_{2.5}$ from all sectors, the $NO_3^-$ share slightly increased from 29.5% to 31.6%, while the $NH_4^+$ share changed little from 12.5% to 12.1%. The sum of SNA share in the shipping-related $PM_{2.5}$ rose from 69.7% to 83.5%, whereas that in the $PM_{2.5}$

from all sectors remained at the same level of ~58%. The $PM_{2.5}$ pollution from shipping which coastal urban areas suffer is mainly caused by the transport and aging processes of pollutants emitted by ships in the atmosphere from water to land, which can explain the higher SNA share in the shipping-related $PM_{2.5}$.

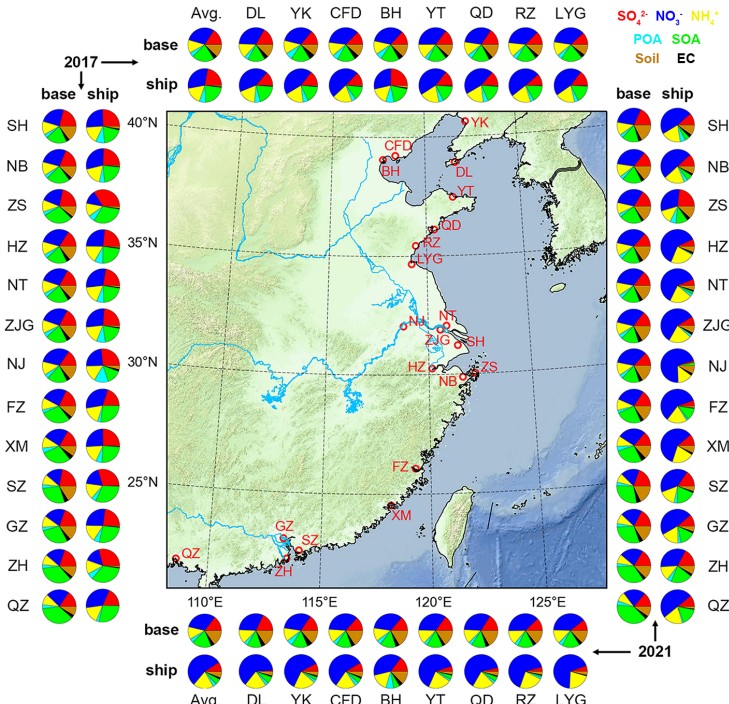

**Figure 14.** Simulated chemical speciation of $PM_{2.5}$ from all sectors (base) and shipping (ship) over the representative main port cities of China in 2017 (left and top) and 2021 (right and bottom). Avg. denotes the annual average. DL, YK, CFD, BH, YT, QD, RZ, LYG, SH, NB, ZS, HZ, NT, ZJG, NJ, FZ, XM, SZ, GZ, ZH, and QZ are the abbreviations of Dalian, Yingkou, Caofeidian, Binhai, Yantai, Qingdao, Rizhao, Lianyungang, Shanghai, Ningbo,
Zhoushan, Hangzhou, Nantong, Zhangjiagang, Nanjing, Fuzhou, Xiamen, Shenzhen, Guangzhou, Zhuhai, and Qinzhou, respectively. The cities in the northern region are placed at the top and bottom, while the cities in the southern region are placed on the left and right.

Regarding the shipping-related $PM_{2.5}$, the reduction in primary $PM_{2.5}$ emissions made SNA more important, with the total SNA share rising from 69.7% to 83.4%. However, the organic aerosol share



decreased from 28.4% to 9.0%. The SOA share reduced from 22.2% to 5.6%, and the decrease in the

SOA share was much more significant than that in the POA share. The EC share increased from 0.8% to

1.8%, corresponding to the increase in the mapping factors of particulate EC (PEC) for ship emissions

from 4.1% to 7.0%. Due to the substantial contribution of SNA, the EC shares in the shipping-related

$PM_{2.5}$ were significantly lower than those in the primary $PM_{2.5}$ from shipping.

At the city level, both the highest $NO_3^-$ and $NH_4^+$ shares in the shipping-related $PM_{2.5}$ in 2017 were

found in Rizhao corresponding to the lowest $SO_4^{2-}$ share. However, these values were obtained in Nanjing

in the inland ultra-low sulfur control areas, with the $NO_3^-$ share up to 70.7%. The $SO_4^{2-}$ shares displayed

lower values in the northern region and higher values in the southern region, which may be partly related

to the lower aerosol acidity in the northern region (Wang et al., 2022). Zhoushan, with low $NH_3$ emissions,

showed the highest $SO_4^{2-}$ shares both in 2017 (33.9%) and 2021 (23.2%).

### 3.5.2 Roles of the meteorological factors

Meteorological factors can affect both physical and chemical processes of atmospheric pollutants.

Considering the complexity of non-linear chemistry discussed above, we focused on the physical aspects

here to clarify the roles of meteorological factors on the spatiotemporal patterns of primary PM from

shipping in the concerned port cities. We utilized V as the tracer since V is the most convincing tracer of

ship emissions and does not participate any chemical process in the model. The impact of temporal

variations of emissions was removed because the monthly ship emissions were evenly allocated by hour

in this study. Although this simplification smoothed out the time series of the impacts of ship emissions,

it contributed to the discussion on the effects of meteorological factors such as wind patterns and the

PBLH. Ship emission inventories will be produced in hourly resolution based on the AIS data and used

in monthly to long-term simulations in our future work.

Figure 15 delineates the diurnal variations of the V concentration from shipping over the port cities in

every season of 2017 when ship emissions dominated the sources of ambient V. The lowest values

occurred at noon in most cases, corresponding to the highest PBLH. However, the time when the highest

values appeared showed significant differences by city, even by season in the same city. If the PBLH

plays a crucial role, the V concentration will simply increase with the reduction of the PBLH and reach

the peak level in early morning. This pattern was found in four cities in the YRD including Shanghai,

Hangzhou, Nantong, and Zhangjiagang in every season. A secondary peak during 20:00−21:00 was



observed in Shanghai in summer, though subtle, which was caused by the intrusion of the sea breeze after
the dissipation of the daytime weak convergence (Shang et al., 2019; Zhai et al., 2023). Shanghai is more
likely to be affected by large synoptic systems, and urbanization results in a weaker land-breeze pattern;
meanwhile, the contribution of inland ship emissions is considerable in downtown Shanghai. The
simulation results confirmed the finding in our previous study conducted in Shanghai based on field
observation (Yu et al., 2021).

In Zhuhai, Shenzhen, and Fuzhou, the V concentrations from shipping peaked before midnight with
values significantly higher than those during early morning in every season, indicating that the sea-land
breeze circulation (SLBC) significantly affected the diurnal patterns in these cities, especially Zhuhai.
The SLBC impact in downtown Guangzhou was much weaker than that in Zhuhai and Shenzhen, which
was due to the dense inland shipping activities in Guangzhou. In Binhai, Caofeidian, and Yingkou, three
ports located in the Bohai Bay, the SLBC impact was significant in at least three seasons. In comparison,
the SLBC impact on the diurnal variations was smaller in Qingdao and Rizhao, two cities adjacent to the
Yellow Sea. Nevertheless, this impact was found in every season; the land-breeze could block the
transport of ship emissions from marine areas and counteract the PBL compression effect. In Dalian and
Yantai, the SLBC impact was noticeable in summer and autumn. The SLBC exerted impact in a certain
season such as summer in Ningbo and Zhoushan while winter in Xiamen. This study adopted the grid
resolution of 9 km and still characterized the impact of the SLBC, a type of local scale system. The results
in this study are rather different from those in a study conducted in the eastern United States with scarce
SLBC impact using the WRF-CAMx (Golbazi and Archer, 2023).

Besides, the seasonal variations of the V concentration from shipping in the port cities are also shown
in Fig. 15. The concentration peaked in spring for most of the cities due to the weak onshore airflows.
Shanghai, Zhoushan, Nantong, Fuzhou, and Xiamen, located along the eastern to southeastern coast
exhibited the lowest levels in autumn. The lowest levels were observed in winter for all the northern
cities affected by the prevailing northwesterly wind as well as two southern cities including Guangzhou
and Shenzhen affected by the prevailing northeasterly wind. The winter monsoon was adverse to the
transport of ship emissions to these cities. In the other cities, poor diffusion conditions enhanced the
wintertime concentration levels. It is of interest that cities close in distance did not always perform the
same seasonal pattern. For example, in the YRD, Ningbo and Zhoushan, adjacent to each other, showed
different seasonal patterns. The southerly winds in July were conducive to the transport of pollutants



emitted by ships in Ningbo Port to the Zhoushan Islands, whereas the northerly and northeasterly airflows

in January and October had opposite effects. In the PRD, Guangzhou and Shenzhen were right located

downwind of dense ship emissions in July when the southerly to southeasterly airflows prevailed, and

thus the V concentration from shipping was at a relatively high level. However, for Zhuhai, located in

the southwest of the PRE, the increase in the V concentration caused by ship emissions was the smallest

in summer. Therefore, in addition to the airflow intensity, the relationship between areas with high ship

emissions and prevailing airflow directions is an important factor affecting the seasonal pattern of the

concentration of primary PM emitted by ships in receptor cities.

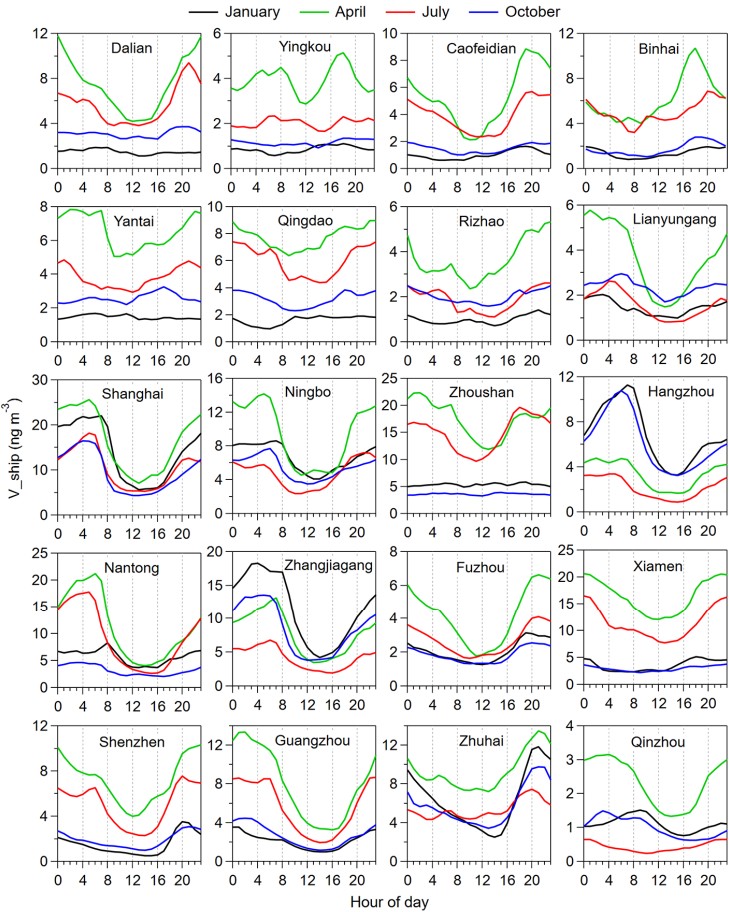

**Figure 15.** Simulated diurnal variations of the V concentrations from shipping (V_ship) over the representative port
cities of China in January, April, July, and October of 2017.





**4 Conclusions**

To meet the requirements of the IMO Regulation, China carried out staged fuel oil policies for sea-going vessels including the DECA 1.0 and the DECA 2.0 which took into effect in 2017 and 2019, respectively. Besides, the inland emission control areas were implemented in 2019 and became more stringent in 2020. It is of significance to evaluate the effects of the low sulfur fuel policies on air quality in China in the context of increasing shipping activities. We updated the ship emission inventory in China to 2021 based on the AIS data. The emissions of V and Ni from shipping were constrained by the field observational data and the results of on-board emission measurements. The WRF/CMAQ model were utilized to simulated the impacts on $PM_{2.5}$ in China as well as its gas precursors ($SO_2$ and $NO_2$) and components from 2017 to 2021 based on the zero-out method. The model reproduced the V and Ni concentrations as well as their seasonal and diurnal variations, whereas underestimated the secondary aerosol concentrations.

In the CECA and inland areas of China, the $SO_2$, $PM_{2.5}$, V and Ni emissions from shipping on monthly averages were 40.9 kt, 7.6 kt, 118.8 t, 41.6 t in 2017, and dropped to 12.9 kt, 5.1 kt, 11.0 t, and 24.1 t in 2021, respectively. The $NO_x$ emissions from shipping increased by 51.8% from 2017 to 2021 due to the increase in ship activities. However, the emissions of $SO_2$, $PM_{2.5}$, V, and Ni from shipping reduced by 68.4%, 32.8%, 90.8%, and 42.0%, respectively, due to the implementation of the IMO Regulation and China's inland river emission control areas.

The $SO_2$ and $NO_2$ concentrations from shipping showed higher values along the main routes and spread to both sides. The maximum $SO_2$ ($NO_2$) concentrations from shipping were 6.1 µg m$^{-3}$ (17.9 µg m$^{-3}$) in 2017 and 2.5 µg m$^{-3}$ (26.0 µg m$^{-3}$) in 2021. In most areas, the peak values occurred in spring while autumn for the valley values, which was attributed to the direction and intensity of the prevailing airflows. The shipping-related $PM_{2.5}$ concentration displayed more uniform spatial patterns. In 2017, the areas with concentrations over 1 µg m$^{-3}$ covered most of the Yellow and Bohai Seas and extended to the Middle Yangtze River, with the maximum of 3.8 µg m$^{-3}$ near Zhoushan Islands. In 2021, these areas shrunk to the coast and the central Yangtze River Basin in 2021, with the maximum of 2.6 µg m$^{-3}$ in the eastern Shandong Peninsula. The seasonal patterns of the shipping-related $PM_{2.5}$ concentration differed by region mainly due to the seasonality of secondary aerosol formation, whereas those of the contributions of ship emissions to was also affected by the East Asia Monsoon considering the impacts of land-based sources.



The shipping-related PM$_{2.5}$ concentration peaked in spring when the SNA formation was significant

under the conditions of weak onshore airflows and abundant gas precursors. The offshore marine areas

suffered from higher SOA concentration related to ship emissions in summer. The northeasterly airflows

diluted the pollution caused by shipping in autumn. NH$_3$ was consumed by land-based emissions in prior

to ship emissions in winter, leading to low shipping-related PM$_{2.5}$ concentration in winter. The springtime

concentrations of shipping-related PM$_{2.5}$, V, and Ni experienced staged reduction in most simulated areas

from 2018 to 2021.

At the city level, the contributions of ship emissions to the PM$_{2.5}$ concentration over China's main port

cities ranged from 3.0% to 17.4% in 2017 and 2.5% to 10.3% in 2021. The change rates of the

concentrations of PM$_{2.5}$, SO$_4^{2-}$, NO$_3^-$, NH$_4^+$, carbonaceous aerosols, V, and Ni related to ship emissions

were -32.7%, -74.0%, +11.0%, -27.5%, -76.9%, -90.3%, and -38.4%, respectively. NO$_3^-$ has become the

dominant species accounting for 54.6% in the shipping-related PM$_{2.5}$ after 2020, and can contribute to

high levels of nocturnal PM$_{2.5}$ concentration. The increasing NO$_x$ emissions from shipping and their

potential impacts on PM$_{2.5}$ and O$_3$ are of concern, which calls for the expansion of the Tier III Regulation

in more coastal waters worldwide. Besides, the sea-land breeze circulation played an important role in

the diurnal patterns of the concentration of primary particulate matter from shipping in most seaports,

while a minor role was found in Shanghai and Guangzhou with large inland ship emissions. Our findings

suggest that it is important to consider both transport pathways and formation mechanisms of secondary

aerosols to combat the PM$_{2.5}$ pollution caused by shipping in different regions.

**Data availability**

The data derived from the ship emission model and the WRF/CMAQ model presented in this paper can

be obtained from Yan Zhang (yan_zhang@fudan.edu.cn) upon request.

**Author contribution**

GY: investigation, methodology, software, validation, formal analysis, data curation, visualization, and

writing – original draft preparation; YZ: conceptualization, investigation, supervision, methodology,

validation, project administration, funding acquisition, and writing – review and editing; QW: validation,

data curation, and writing – review; ZH: methodology, software, and data curation; SJ: methodology,



software, and data curation; FY: data curation, funding acquisition, and writing – review; XY: writing – review and editing; CH: supervision, data curation, and writing – review and editing

**Competing interests**

The authors declare that they have no conflict of interest.

**Acknowledgements**

The work was supported by the National Natural Science Foundation of China (No. 42077195) and the Natural Science Foundation of Shanghai Committee of Science and Technology, China (No. 22ZR1407700).

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
