# Peer review of "Changes in the impacts of ship emissions on PM2.5 and its components in China under the staged fuel oil policies"

_EGUsphere, 2024_

## Author Response (AR1)

**Response to Referees**

We sincerely thank the valuable comments and suggestions from the referees. We extensively revised the manuscript according to the referees' comments. Below we list our point-by-point replies to the comments and the descriptions of the changes we made in the revised manuscript.

**Referee #1**

Shipping emissions play important roles in coastal air quality, influencing the ambient concentrations of $SO_x$, $NO_x$, $PM_{2.5}$ and their components in atmospheric environment. Recent years, ship fuel oils have experiencing multi-stages transitions due to low sulfur fuel oil policies implemented in national scale and global scales. This study presents comprehensive impacts of changes in ship emissions on $PM_{2.5}$ as well as its gas precursors and primary and secondary components in China. The emission inventory updating based on field and on-board measurements is original and crucial for estimating atmospheric influence under different policy stages. Generally, this paper was structured and written well. The study results could add inputs for providing good reference for air pollution prevention in coastal cities.

There are also some places need to be clarified clearer. I just post my suggestions and my questions as the below:

(1) Section Introduction: the introduction should be updated to include more recently published research results, especially introduce some new research results after IMO 2020 policy.

**Response:** We thank the referee for this comment. Four new references in which data covered the years since 2020 were added in the Introduction and are listed as follows:

1. Luo, Z., Lv, Z., Zhao, J., Sun, H., He, T., Yi, W., Zhang, Z., He, K., and Liu, H.: Shipping-related pollution decreased but mortality increased in Chinese port cities, Nature Cities, 1, 295-304, https://doi.org/10.1038/s44284-024-00050-8, 2024.

2. Song, S.-K., Shon, Z.-H., Moon, S.-H., Lee, T.-H., Kim, H.-S., Kang, S.-H., Park, G.-H., and Yoo, E.-C.: Impact of international Maritime Organization 2020 sulfur content regulations on port air quality at international hub port, Journal of Cleaner Production, 347, https://doi.org/10.1016/j.jclepro.2022.131298, 2022.

3. Wang, X., Liu, H., Zhang, J., Fu, X., Chen, D., Zhang, W., Yi, W., Lv, Z., Zhang, Q., and He, K.: Global shipping emissions from 1970 to 2021: Structural and spatial change driven by trade dynamics, One Earth, https://doi.org/10.1016/j.oneear.2025.101243, 2025.

4. Yi, W., Wang, X., He, T., Liu, H., Luo, Z., Lv, Z., and He, K.: The high-resolution global shipping emission inventory by the Shipping Emission Inventory Model (SEIM), Earth System Science Data, 17, 277-292, https://doi.org/10.5194/essd-17-277-2025, 2025.

(2)  Section Methodology: The authors stated that the emission updating was based on previous measurement study in Shanghai. Since the following simulation work cover the national scale, how about the components changing situation of $PM_{2.5}$ in other coastal cities? Have any results been reported that could be compared with each other?

**Response:** We sincerely thank the referee for this concern. The online data of ambient $PM_{2.5}$ components in Shanghai in this paper were achieved from Shanghai Environmental Monitoring Center, one of the authors' affiliations. We agree that the $PM_{2.5}$ component data in other China's coastal cities can help to evaluate the model performance on a larger scale. However, we are sorry that online data of this type are not open-access currently, and especially acquisition of multiple-year data need extensive cooperation with institutes in other coastal cities. Therefore, in this paper, on an hourly scale, we only used open-access hourly measurement data of $PM_{2.5}$, $SO_2$, $NO_2$, and $O_3$ for model evaluation in the cities except for Shanghai.

It is noted that the emissions of trace elements from shipping (V and Ni) were updated based on the previous measurement study in Shanghai, which is one of the highlights of this study. In the revised supplement, we reviewed recently published studies, and compared the changes in the average ambient V and Ni concentrations in several coastal cities including Qingdao, Jiaxing, Xiamen, Guangzhou, and Seoul in the model domain since the DECA 1.0 period to the simulation results (Table S13). The observation periods in other studies could not match up with the simulation periods in this study very well, and thus the general trends were compared. The comparison results can be found in the last part of Text S3 in the supplement. After updating the V and Ni emissions from shipping, the model could generally reproduce the decreasing trends of ambient V and Ni concentrations in China's coastal areas since 2017, not only in Shanghai.

**Table S13.** Comparison of the observed and simulated V and Ni concentrations in the model domain (unit: ng m$^{-3}$).

| City | Observation | | | Simulation (this study) | | | Reference |
|------|-------------|---|---|-------------------------|---|---|-----------|
| | DECA 1.0 | DECA 2.0 | IMO 2020 | DECA 1.0 | DECA 2.0** | IMO 2020 | |
| Shanghai | V: 11.31 | V: 7.23 | V: 1.36 | V: 13.4 | V: 7.7 | V: 1.7 | This study |
| | Ni: 5.32 | Ni: 5.39 | Ni: 3.52 | Ni: 5.6 | Ni: 4.2 | Ni: 3.2 | |
| Qingdao | V*: 4.01 | V*: 1.50 | V*: 0.84 | V: 5.9 | V: 6.9 | V: 1.6 | Du et al. |
| | Ni*: 3.03 | Ni*: 2.13 | Ni*: 3.55 | Ni: 4.0 | Ni: 3.8 | Ni: 3.1 | (2024) |
| Jiaxing | V: 5.52 | V: 4.29 | V: 1.34 | V: 8.3 | V: 5.4 | V: 1.2 | Zhang et al. |
| | Ni: 4.68 | Ni: 4.44 | Ni: 4.18 | Ni: 3.9 | Ni: 3.2 | Ni: 2.2 | (2024a) |
| Xiamen | V: 7.35 | | V: 1.60 | V: 9.0 | | V: 1.2 | Li et al. |
| | Ni: 3.22 | | Ni: 1.87 | Ni: 3.7 | | Ni: 2.3 | (2025) |
| Guangzhou | | V: 1.1 | | | | V: 1.3 | Zhang et al. |
| | | Ni: 3.1 | | | | Ni: 2.6 | (2024c) |
| Seoul | V: 4.6 | V: 2.1 | V: 0.4 | V: 2.3 | V: 3.2 | V: 0.6 | Yeo et al. |
| | Ni: 1.7 | Ni: 0.8 | Ni: 0.5 | Ni: 1.5 | Ni: 1.7 | Ni: 1.7 | (2024) |

Note: * The observed V and Ni concentrations in Qingdao are PM$_1$-bound. The other data are PM$_{2.5}$-bound.

** The simulated month for the DECA 2.0 is April 2019 with peak concentrations in a year.

(3) Model Performance: Figure 2(b), there was not the evident reduction in the PM$_{2.5}$ concentration contributed by ship in 2019 compared with 2017 and 2018. Since the policy is step by step as stated in Line84-88 in Introduction, why didn't PM$_{2.5}$ response well with the staged policy?

**Response:** We thank the referee for this comment. It is interesting that the shipping-related PM$_{2.5}$ concentrations in the coastal port cities did not exhibit an evident reduction in 2019 compared to 2017 and 2018. This result has been discussed in the last paragraph of Sect. 3.4.1 in the original manuscript:

*"In April 2019, high values were closer to the coastline compared to the DECA 1.0 period, and the CECA border was indistinct compared to the pattern of the SO$_2$ concentration from shipping. This result highlights the crucial role of secondary aerosols in the shipping-related PM$_{2.5}$ in the presence of plenty NH$_3$ emissions. The highest concentration reduced to 6.5 μg m$^{-3}$, whereas there was an increase along the coast of Shandong due to more secondary aerosol formation."*

In our previous study based on field measurement and source apportionment in Shanghai, we found that the shipping-related PM$_{2.5}$ concentration even increased from 2.0 μg m$^{-3}$ in 2018 to 3.1 μg m$^{-3}$ in 2019, which was likely due to the transport of aged particles outside the CECA and the increasing nitrate formation. The simulation results in this study generally matched the observational results.

Considering that Sect. 3.2 focuses on the model performance and uncertainties, the nonlinear reduction

effect of shipping-related $PM_{2.5}$ is discussed in Sect. 3.4.1. We added a note in Sect. 3.2 as follows:

*"It is noted that the shipping-related $PM_{2.5}$ concentration did not show significant reduction in 2019 compared to 2017 and 2018, which was likely due to the increase in the impact of secondary aerosols (Fig. 2b). This result will be discussed further in Sect. 3.4.1."*

In addition, we revised the discussion in the last paragraph in Sect. 3.4.1 as follows:

*"In $NH_3$-rich port areas, the increasing $NO_x$ emissions from shipping resulted in more nitrate formation. Meanwhile, the reduction of $SO_2$ emissions provided more opportunities for $HNO_3$ to react with $NH_3$. Besides, the difference between the shipping-related $PM_{2.5}$ concentrations inside and outside the CECA border was much smaller compared to the $SO_2$ concentrations from shipping, which was due to the onshore transport of aged aerosols from marine areas outside the CECA where high-sulfur fuel oils were still used. These aged aerosols could also contribute to the shipping-related $PM_{2.5}$ in the port cities. Therefore, the impact of secondary aerosols partly offset the effect of primary PM emission reduction during the DECA 2.0 period."*

(4)     Results: As authors stated as "$NH_3$ is sufficiently consumed by $SO_2$ and $NO_x$ from land-based emissions, and thus the formation of secondary aerosols related to shipping is inhibited, which called the competitive mechanism by land-based sources.", what is the change in the competitive relationship between $SO_2$ and $NO_x$ in marine atmosphere after low sulfur fuel oil policy?

**Response:** We thank the referee for this comment. We proposed the competitive mechanism by land-based sources to explain the result that the wintertime concentrations of the shipping-related $PM_{2.5}$ were very low in both coastal and offshore marine areas. This competitive mechanism results from the competition of consuming $NH_3$ between land-based sources and shipping, not the competition between $H_2SO_4$ and $HNO_3$ reacting with $NH_3$. It is apparent that the reduction in $SO_2$ emissions can give more opportunities for $HNO_3$ to react with $NH_3$ after 2020 compared to 2017, which was analyzed in Sect. 3.4.3. **To avoid confusion, we change "*the competitive mechanism by land-based sources*" to "*the domination effect of land-based sources*".**

The sentence pointed out by the referee is to discuss the seasonal patterns of the concentrations of shipping-related $PM_{2.5}$. In winter, the gas precursors of secondary inorganic aerosols, i.e., $SO_2$, $NO_2$, and $NH_3$ accumulate and can generate high $PM_{2.5}$ concentrations on coastal land areas. And, marine atmosphere is $NH_3$-poor as there are very few $NH_3$ emissions in marine areas (Fig. S11). Most $NH_3$ over

coastal waters is related to the transport from land areas, and is reacted by $SO_2$ and $NO_2$ from land-based sources in prior to those from shipping. As a result, there is insufficient $NH_3$ to react with $SO_2$ and $NO_x$ emitted by ships. This phenomenon can be found in winter of both 2017 and 2021. Underlining the seasonal patterns, we used different scales of the color bars (Fig. 6 in the revised manuscript). Here, we presented another version (Fig. R1) with the same scale for 2017 and 2021. There was no evident change in the spatial distribution of the shipping-related $PM_{2.5}$ concentrations in winter from 2017 to 2021.

[Figure]

**Figure R1.** Seasonal patterns of the potential impacts of ship emissions on $PM_{2.5}$ ($PM_{2.5\_}$ship) concentrations in (a) January, (b) April, (c) July, and (d) October of 2017; and (e) January, (f) April, (g) July, and (h) October of 2021.**The color bars are in the same scale.**

(5)      Minor suggestions: maybe move some figures and text into the supplementary material to make the full length shorten.

**Response:** We sincerely thank the referee for this suggestion. We comprehensively considered the suggestions of the two referees: some text of Sect. 3.2 (model performance) was moved to the supplementary material; the figure number was reduced from 15 to 12; some location-specific results in Sect. 3.3&3.4 were streamlined; and the conclusion was shortened. The word count from the abstract to the conclusions was reduced by 1546.

**Referee #2**

This study investigates the impacts of low-sulfur fuel policies on $PM_{2.5}$, its gaseous precursors ($SO_2$ and $NO_2$), and its components (V, Ni, $SO_4^{2-}$, $NO_3^-$, $NH_4^+$, etc.) using WRF-CMAQ model simulations with an updated shipping emission inventory for China from 2017 to 2021. The results effectively quantify the effects on each component, compare air pollution variations in port cities across China, and examine potential meteorological influences. These findings provide timely and valuable insights for air quality studies in coastal regions. Additionally, the manuscript is well-structured and detailed. However, several issues need to be clarified before publication. Below are specific comments:

Specific Comments

Lines 80–105: In response to IMO regulations, China has implemented various regional strategies in recent years. To improve clarity, the authors should include a table summarizing key aspects of China's IMO regulations, such as revisions and objectives. Additionally, the full names of "IMO," "Tier II," and "Tier III" should be provided for clarity.

**Response:** We thank the referee for this comment. We added a table summarizing the staged sulfur regulations in China in the revised supplement (Table S1). We provided the full name of "IMO", i.e., the International Maritime Organization, in Line 90 in the revised manuscripts. However, no full name is available for Tier II or Tier III. We added the $NO_x$ emission factor limits by the Tier III (Line 87) and the Tier II (Line 102−103) in the revised manuscripts for clarity.

Section 2 (Methods): The authors cite previous studies to describe updates to the emissions inventory and CMAQ model. However, all details directly relevant to this manuscript's results should be explicitly presented, even if they have been previously published.

**Response:** We thank the referee for this comment. We have already presented detailed information on the configuration of WRF and CMAQ, the emission factor settling of pollutants emitted by ships, the sources of land-based, the mapping of $PM_{2.5}$ and NMVOCs into AERO7 and CB6 respectively, and the vertical profiles of emission inventories. Only the CMAQ code revision for V and Ni and the setup of the ship emission model were referred to our previous studies. We added these two parts in the revised supplement. The information on the CMAQ code revision for V and Ni are shown in Text S1. The technical details on the setup of the ship emission model can be found in Text S2.

Lines 215–225: It is recommended to mark recurring areas (e.g., Yangtze River Delta, Pearl River Delta) on a map. For the 21 port cities (DL, YK, CFD, BH, YT, QD, RZ, LYG, SH, NB, ZS, HZ, NT, ZJG, NJ, FZ, XM, SZ, GZ, ZH, and QZ), abbreviations should be used consistently in both the text and figures.

**Response:** We thank the referee for this recommendation. We labelled the Yangtze River Delta (YRD) and the Pearl River Delta (PRD) in Fig. 1. And, we used the abbreviations of the port cities consistently in both the text and figures.

Line 235:

(a) Table A1 is not found in the manuscript.

**Response (a):** We thank the referee for pointing out this mistake. The original Table A1 was moved to the supplement before we submitted the paper. Its current number is Table S7. We labelled the Yangtze River Delta (YRD) and the Pearl River Delta (PRD) in Fig .1.

(b) The values of IoA and RMSE were used to assess the simulation performance of V and Ni, while different statistical indicators (e.g., r and NMB) were used for $SO_2$, $NO_2$, MDA8 $O_3$, $PM_{2.5}$, $SO_4^{2-}$, $NO_3^-$, and $NH_4^+$ (Tables S8 & S9). Is there a specific reason for this distinction?

**Response (b):** We thank the referee for the comment. We agree that the statistical indicators should be consistent for all the concerned pollutants. We added the RMSE and IoA values, and removed mean values of observational and simulation data in Table S9 and Table S10 in the revised supplement. And, we added the IoA values for SNA in Table S11, as well as the r and NMB values for V and Ni in Table S12 in the revised supplement. It is noted that the r values of the $PM_{2.5}$ components were relatively lower compared to main pollutants such as $SO_2$, $NO_2$, and $PM_{2.5}$. The variations of SNA are related to complex physical and chemical processes. Reproducing the time series of shipping tracers (V and Ni) is more difficult considering their rather low concentration levels and large uncertainties due to the diversity of fuel oils. In addition, the observational data are based on one supersite in Shanghai where large local emissions can affect the temporal variations of these chemical species. In this paper, if monthly NMB values are within the range of ±50% or monthly IoA values are above 0.50, the model performance can be considered acceptable.

Lines 250–260: Some data in the manuscript differs from Table 2. For example: "The monthly average V emissions from shipping dropped by 90.8%, from 118.8 t in 2017 to 43.9 t in 2021." "The monthly

average Ni emissions decreased from 11.0 t in 2017 to 24.1 t in 2021, a reduction of 42.0%." Please verify and ensure consistency.

**Response:** We thank the referee for pointing out these mistakes. The data in Table 2 are correct, but some data in the text were mistaken. These two sentences were revised as follows:

"*In addition, the monthly average V emissions from shipping experienced a dramatic drop (by 90.8%) from 118.8 t in 2017 to 11.0 t in 2021. The monthly average Ni emissions decreased from 41.6 t in 2017 to 24.1 t in 2021, with a reduction of 42.0%.*"

Section 3.2 (Model Performance): Some content could be moved to the supplementary material for conciseness.

**Response:** We thank the referee for the comment. We moved some text to the supplement. The word count of Sect. 3.2 was reduced from 834 to 520.

Sections 3.3 and 3.4: Since policy changes were implemented gradually, the differences in air pollution trends due to policy shifts should be highlighted more clearly. Meanwhile, location-specific results could be streamlined to shorten the manuscript.

**Response:** We appreciate the referee for the comment. To keep to "the staged fuel oil policies" in the title, it is better to show the changes in the impacts of ship emissions stage by stage. We completely reorganized Sect. 3.3 and Sect. 3.4, replaced the figures, and highlighted the effects of policy shifts more clearly. We focused on interannual variations and provided the change rates due to each policy shift. Besides, some location-specific results were streamlined. The total word count of these two sections has been reduced by 983. The results and discussion on the effects of the two-stage policy shifts for every pollutant (species) still take up some space. From our perspective, these two sections become more closely to the title and easier to read.

Lines 370–375:

(a) The maximum $SO_2$ concentration in 2018 reached 30.1 μg/m$^3$, significantly higher than in other years. What could be the underlying cause?

**Response:** We thank the referee for the comment. In the original manuscript, we used the maximum grid values in the Domain02 to describe the annual variations of the air pollutant concentrations. However,

some maximum values are unexpectedly high. The concentrations of V and Ni contributed by shipping in 2018 were also significantly higher than those in the other years, which was consistent with the pattern of $SO_2$. The maximum value of shipping-related $NO_2$ concentration increased from 25.3 μg m$^{-3}$ in 2017 to 48.5 μg m$^{-3}$ in 2018, with an increase rate of 92%. As shown in Table 2, the total $SO_2$ and $NO_x$ emissions from shipping in April 2018 increased by 42% and 37% compared to April 2017 respectively, and the increase rates were significantly lower compared to the maximum concentration values. This rapid growth of shipping activities was found both in the coastal YRD and PRD, and its spatial distribution was uneven. The remarkable increase in ship emissions and pollutant concentrations in individual grids could not represent the overall trend on a national scale, which should be regarded as a kind of outlier. These outliers at a very local scale were authentic, but could not contribute to the discussion.

To avoid the interference of the very high values on a local scale, in the revised manuscripts, we changed the maximum values to the 99$^{th}$ percentile (P$_{99}$) values for the annual variations of concentrations in Sect. 3.3. The Domain02 has 215×233 grids, and many of them have no ship emissions. The P$_{99}$ value excluded 500 grids with very high concentration values. As expected, using P$_{99}$ can help present the overall trend from 2017 to 2021. The discussion on the interannual variations of the $SO_2$ concentrations from shipping was revised as follows:

"*The $SO_2$ concentrations from shipping experienced evident staged reduction. The P$_{99}$ values from 2017 to 2021 were 4.1, 5.2, 2.6, 1.5, and 1.2 μg m$^{-3}$ in the chronological order. Comparing the concentrations in 2018 and 2019, it was found that the DECA 2.0 had the effect of halving the $SO_2$ concentrations from shipping. After 2020, hotspots with values over 4 μg m$^{-3}$ were only found along the coast of Zhejiang with dense fishing activities. The P$_{99}$ value in 2021 was even lower than that in 2020, which was related to the rebounce of fishing activities after the COVID-19 lockdown in early 2020. Comparing the P$_{99}$ values in 2019 and 2021, a reduction rate of 53.8% was obtained due to the IMO Regulation and was comparable with that brought by the DECA 2.0.*"

(b) A high-pressure system typically leads to stable weather with low wind speeds, facilitating pollutant accumulation.

**Response:** We thank the referee for the comment. There was a divergent wind field at the surface level in the Yellow Sea in April 2020. Usually, a divergent wind field is brought by a high-pressure system. This wind field pattern can help the horizontal diffusion of pollutants at the surface level. However, a

convergent wind field brought by a low-pressure system can lead to the pollution accumulation near the convergence center.

$SO_2$ concentrations are mainly affected by emissions and meteorology as $SO_2$ is a kind of primary pollutant. To explain the interannual variation, we should first examine the variation of emissions and then that of meteorological factors. We plotted the interannual variations of $SO_2$ and $NO_x$ emissions from shipping in Fig. S1 in the revised supplement. Comparing Fig. S1d and Fig. S1e, it was found that in the Yellow Sea, the $SO_2$ emission intensity along the long shipping routes between China and Korea as well as between northern and southern China in April 2020 were lower than those in April 2021, which was due to the impact of the COVID-19 lockdown on long-distance shipping. We modified the discussion on the variation of the $SO_2$ concentration from shipping accordingly in Line 357−360 in the revised manuscript.

[Figure]

**Figure S1.** Spatial distributions of **(a–e)** the $SO_2$ emissions from shipping ($SO_2$_ship) and **(f–j)** the $NO_x$ emissions from shipping ($NO_x$_ship) in Domain 2 of the CMAQ model in April from 2017 to 2021 in the chronological order from left to right.

Line 415: The discussion appears unclear, as contributions from sea salt are presumably excluded in Figure 5.

**Response:** We thank the referee for this comment. Beginning with version 4.5, CMAQ has included online calculation of sea-salt emissions. The inline sea salt emission module was also enabled in the CMAQ v5.4 applied in this paper. Sodium (Na) is one of the tracers of sea salt. The spatial distributions of the Na concentrations in $PM_{2.5}$ are shown in Fig. S7 in the revised supplement. The Na concentrations over marine areas were higher than those over land areas, and their spatial distribution patterns in 2017

and 2021 were different, which indicated that the inline sea salt emissions were included in the model. Therefore, the lower contributions of ship emissions to $PM_{2.5}$ concentrations in remote marine areas were related to the contributions of sea salt.

[Figure]

**Figure S7.** Spatial distributions of the sodium (Na) concentrations in $PM_{2.5}$ in (a) 2017 and (b) 2021.

Lines 430–435: Regarding the seasonal shift in high $PM_{2.5}$ concentrations from southern to northern regions, the impact of the Asian Summer Monsoon should be considered.

**Response:** We sincerely thank the referee for the comment. The East Asian Monsoon is one of the factors impacting the seasonal patterns of pollutant concentrations. As is known, the $PM_{2.5}$ concentrations over China's land areas are lower in summer compare to the other seasons. The summer monsoon brings relatively clean air mass from marine areas and the diffusion conditions are better compared to the other seasons. However, in comparison, the seasonal pattern of the shipping-related $PM_{2.5}$ concentrations over China's coastal areas was rather different. Secondary aerosols constitute a substantial portion of shipping-related $PM_{2.5}$. This fact resulted in the difference between the seasonal patterns of primary pollutants (e.g., $SO_2$, V, and Ni) and those of secondary pollutants.

As discussed in Sect. 3.3, the $SO_2$ concentrations were lower in summer than those in spring due to better diffusion conditions. However, the spatial distributions in spring and summer were similar (Fig. S5). Meanwhile, the seasonal shift from southern to northern regions from winter to summer was not found. If this seasonal shift was driven by the Asian Summer Monsoon, it should be found both in the concentrations of $SO_2$ and $PM_{2.5}$ contributed by ship emissions. However, this seasonal shift was found in $PM_{2.5}$ only and was attributed to secondary aerosol formation. In summer, as discussed in Sect. 3.4.3, high temperature and good lighting conditions are in favor of BVOC emissions and photochemical

reactions, resulting in abundant oxidants in the background atmosphere to generate $O_3$, sulfate, and SOA (Fig. S12). Besides, the $NH_3$ emissions were also higher in summer and provided an $NH_3$-rich condition to form SNA (Fig. S11). In winter, as discussed in Sect. 3.4.1 and Sect. 3.4.3, the coastal marine areas were in oxidant-poor and $NH_3$-poor conditions, resulting in low concentrations of shipping-related $PM_{2.5}$. Therefore, chemistry played a much more important role in the seasonal shift of shipping-related $PM_{2.5}$ concentrations compared to the East Asian Monsoon.

Lines 475–480: Distributions of $NH_3$ emissions from 2017 to 2019 should be provided.

**Response:** We sincerely thank the referee for this comment. As mentioned in Sect. 2.2.2, the $NH_3$ emissions in the MEIC were replaced by the PKU-$NH_3$ inventory in 2017. We did not use the MEIC $NH_3$ emission inventory because $NH_3$ emissions in urban areas may be underestimated and its spatial resolution is coarser (0.25°) compared to the PKU-$NH_3$ (0.1°). However, the latest year of PKU-$NH_3$ is 2017 when the simulation was conducted for this study. Most of $NH_3$ (nearly 90%) is related to agriculture and the emission control measures of $NH_3$ are not stringent as those of other air pollutants in China, leading to a smaller reduction rate for $NH_3$ emissions. Also, this study mainly focused on the impacts of ship emissions on air quality under highly dynamic ship fuel oil policies. So, we just adopted the 2017 PKU-$NH_3$ inventory here for our simulations. The $NH_3$ emission inventory used in this study did not have interannual variations but have monthly variations. To help elucidate the seasonal pattern of the shipping-related $PM_{2.5}$ concentrations, the seasonal pattern of the $NH_3$ emissions was added to the supplement as Fig. S11.

[Figure]

**Figure S11.** Spatial distributions of the NH₃ emissions for the base runs in **(a)** January, **(b)** April, **(c)** July, and **(d)** October of 2017.

Section 3.5: Only PM$_{2.5}$ components in Shanghai were verified. Are similar data available for other port cities?

**Response:** We sincerely thank the referee for this concern. The online data of ambient PM$_{2.5}$ components in Shanghai in this paper were achieved from Shanghai Environmental Monitoring Center, one of the authors' affiliations. We agree that the PM$_{2.5}$ component data in other China's coastal cities can help to evaluate the model performance on a larger scale. However, we are sorry that online data of this type are not open-access currently, and especially acquisition of multiple-year data need extensive cooperation with institutes in other coastal cities. Therefore, in this paper, on an hourly scale, we only used open-access hourly measurement data of PM$_{2.5}$, SO$_2$, NO$_2$, and O$_3$ for model evaluation in the cities except for Shanghai.

It is noted that the emissions of trace elements from shipping (V and Ni) were updated based on the previous measurement study in Shanghai, which is one of the highlights of this study. In the revised supplement, we reviewed recently published studies, and compared the changes in the average ambient V and Ni concentrations in several coastal cities including Qingdao, Jiaxing, Xiamen, Guangzhou, and Seoul in the model domain since the DECA 1.0 period to the simulation results (Table S13). The

observation periods in other studies could not match up with the simulation periods in this study very well, and thus the general trends were compared. The comparison results can be found in the last part of Text S3 in the supplement. After updating the V and Ni emissions from shipping, the model could generally reproduce the decreasing trends of ambient V and Ni concentrations in China's coastal areas since 2017, not only in Shanghai.

**Table S13.** Comparison of the observed and simulated V and Ni concentrations in the model domain (unit: ng m$^{-3}$).

| City | Observation | | | Simulation (this study) | | | Reference |
|------|-------------|------|----------|-------------------------|------------|----------|-----------|
| | DECA 1.0 | DECA 2.0 | IMO 2020 | DECA 1.0 | DECA 2.0** | IMO 2020 | |
| Shanghai | V: 11.31 | V: 7.23 | V: 1.36 | V: 13.4 | V: 7.7 | V: 1.7 | This study |
| | Ni: 5.32 | Ni: 5.39 | Ni: 3.52 | Ni: 5.6 | Ni: 4.2 | Ni: 3.2 | |
| Qingdao | V*: 4.01 | V*: 1.50 | V*: 0.84 | V: 5.9 | V: 6.9 | V: 1.6 | Du et al. |
| | Ni*: 3.03 | Ni*: 2.13 | Ni*: 3.55 | Ni: 4.0 | Ni: 3.8 | Ni: 3.1 | (2024) |
| Jiaxing | V: 5.52 | V: 4.29 | V: 1.34 | V: 8.3 | V: 5.4 | V: 1.2 | Zhang et al. |
| | Ni: 4.68 | Ni: 4.44 | Ni: 4.18 | Ni: 3.9 | Ni: 3.2 | Ni: 2.2 | (2024a) |
| Xiamen | V: 7.35 | | V: 1.60 | V: 9.0 | | V: 1.2 | Li et al. |
| | Ni: 3.22 | | Ni: 1.87 | Ni: 3.7 | | Ni: 2.3 | (2025) |
| Guangzhou | | V: 1.1 | | | | V: 1.3 | Zhang et al. |
| | | Ni: 3.1 | | | | Ni: 2.6 | (2024c) |
| Seoul | V: 4.6 | V: 2.1 | V: 0.4 | V: 2.3 | V: 3.2 | V: 0.6 | Yeo et al. |
| | Ni: 1.7 | Ni: 0.8 | Ni: 0.5 | Ni: 1.5 | Ni: 1.7 | Ni: 1.7 | (2024) |

Note: * The observed V and Ni concentrations in Qingdao are PM$_1$-bound. The other data are PM$_{2.5}$-bound.

** The simulated month for the DECA 2.0 is April 2019 with peak concentrations in a year.

Figure 13:

a) Are the data averaged across all ports? If so, confidence intervals should be provided in the text.

**Response:** We appreciate the referee for this comment. The data were averaged across all ports. As recommended by the referee, the 95% confidence intervals (CIs) of concentrations and contributions were provided in Sect. 3.5.1 of the revised manuscript. The contribution of the secondary SO$_4^{2-}$ to the total SO$_4^{2-}$ related to shipping was recalculated based on port average.

b) Previous discussions indicated an increase in NO$_x$ from shipping. However, Figure 13b shows that the average NO$_2$ level in 2021 was lower than in 2017, which appears inconsistent. Please clarify.

**Response:** We sincerely thank the referee for this comment. Actually, the average NO$_2$ level in 2021 was lower than that in 2017. We realized that the double axis could be misleading. The concentration boxes

and the left y-axis are colored in red, while the contribution boxes and the right y-axis are colored in blue. The x-axis represents the years. The concentration and contribution boxes are grouped by year, and the two groups are split by grey lines. The corresponding sentence in the figure caption was revised as follows: "*The concentration boxes are colored in red corresponding to the left axes, while the contribution boxes are colored in blue corresponding to the right axes.*"

Conclusion: The conclusion should be more concise.

**Response:** We thank the referee for this comment. We abstracted the key points of each section in the revised conclusion. The original conclusion has 732 words, while the revised one has 554 words. Considering the overall length of the main text, the proportion of the conclusions is within an acceptable range (~5%).

Technical Corrections

Lines 125–130: Clarify whether the time resolution is "hourly" or "6-hourly."

**Response:** We thank the referee for this comment. The ECMWF provides the ERA5 reanalysis data with hourly resolution, and this kind of data were used to drive the WRF model in this paper. Thus, "hourly" is correct here. Another kind of meteorological data which are commonly used is the Final Operational Global Analysis Data (FNL) from the National Centers for Environmental Prediction (NCEP) with 6-hourly resolution. At present, it is uncertain how fine the temporal resolution of forced data will obtain better simulation results. At present, there is not enough evidence that using forced data with a certain time resolution can obtain better simulation results. We just kept the original temporal resolution of the ERA5 data.

Lines 160–165: Modify to: "sulfur dioxide ($SO_2$), carbon monoxide (CO), nonmethane volatile organic compounds (NMVOCs)…" The abbreviation should follow its full name upon first mention.

**Response:** We thank the referee for this correction. Now the abbreviations follow their full names upon first mention in the revised manuscript.

Line 760: "Fig. 15" should be "Figure 15."

**Response:** We thank the referee for this comment. However, in the ACP submission guideline, it is noted

as follows: The abbreviation "Fig." should be used when it appears in running text and should be followed by a number unless it comes at the beginning of a sentence, e.g.: "The results are depicted in Fig. 5. Figure 9 reveals that...". Please refer to the website https://www.atmospheric-chemistry-and-physics.net/submission.html.

The manuscript contains minor spelling and grammatical errors that should be corrected.

**Response:** We thank the referee for this comment. We reexamined spelling and grammatical errors throughout the manuscript.